# Subcellular analysis of blood-brain barrier function by micro-impalement of vessels in acute brain slices

Amira Sayed Hanafy [1,2,6], Pia Steinlein[1,2,6], Julika Pitsch [3,4], Mariella Hurtado Silva [5], Natascha Vana[1], Albert J. Becker[3], Mark Evan Graham [5], Susanne Schoch [3], Alf Lamprecht [2] ✉ & Dirk Dietrich [1] ✉

The blood-brain barrier (BBB) is a tightly and actively regulated vascular barrier. Answering fundamental biological and translational questions about the BBB with currently available approaches is hampered by a trade-off between accessibility and biological validity. We report an approach combining micropipette-based local perfusion of capillaries in acute brain slices with multiphoton microscopy. Micro-perfusion offers control over the luminal solution and allows application of molecules and drug delivery systems, whereas the bath solution defines the extracellular milieu in the brain parenchyma. Here we show, that this combination allows monitoring of BBB transport at the cellular level, visualization of BBB permeation of cells and molecules in real-time and resolves subcellular details of the neurovascular unit. In combination with electrophysiology, it permits comparison of drug effects on neuronal activity following luminal versus parenchymal application. We further apply micro-perfusion to the human and mouse BBB of epileptic hippocampi highlighting its utility for translational research and analysis of therapeutic strategies.

In vertebrates, the vascular system constitutes the main pathway to metabolically connect organs throughout the body by delivering nutrients and by collecting and transporting metabolites to the excreting organs via the transport medium blood[1]. The vascular system is enclosed by a continuous cell-based barrier surrounding its lumen. This barrier not only assures an efficient stream of blood but also strongly restricts the free exchange of molecules between blood and tissue and thereby compartmentalizes the milieu of the vascular system from other tissues of the body[2]. However, rather than being generally impermeable, the barrier selectively and actively regulates the passage of molecules in both directions to meet the regional requirements of the nurtured tissue[2].

The brain is the organ with the most tightly and actively regulated vascular barrier called the blood-brain-barrier (BBB)[3]. This likely reflects the brain's strict requirements for a constant milieu of electrolytes and metabolites, despite strongly varying states of neuronal activity, and a partial separation of the brain's immune system from the rest of the body[4]. Maintenance and development of the BBB is regulated by an interaction of three types of cells, endothelial cells (ECs), pericytes (PCs) and astrocytes, which are together with neurons referred to as the neurovascular unit (NVU)[3]. ECs in the brain contiguously border the vascular lumen and are intricately linked to each other by tight junctions, such that they form the primary barrier, which cannot be crossed by most small molecules and water[5]. PCs form a

[1]Department of Neurosurgery, University Hospital Bonn, Bonn, Germany. [2]Department of Pharmaceutics, Institute of Pharmacy, University of Bonn, Bonn, Germany. [3]Section for Translational Epilepsy Research, Dept. of Neuropathology, University Hospital Bonn, Bonn, Germany. [4]Department of Epileptology, University Hospital Bonn, Bonn, Germany. [5]Synapse Proteomics, Children's Medical Research Institute, The University of Sydney, Sydney, Australia. [6]These authors contributed equally: Amira Sayed Hanafy, Pia Steinlein. ✉e-mail: alf.lamprecht@uni-bonn.de; dirk.dietrich@uni-bonn.de

loose mesh on top of ECs and regulate outgrowth of vessels and the development of the BBB[6]. Astrocytes regularly but not contiguously touch ECs with their endfeet while the remainder of their cellular processes extend into the brain parenchyma. Astrocytes sense neuronal activity, have a homeostatic function in the brain parenchyma and regulate the capillary diameter in response to neuronal activity[7].

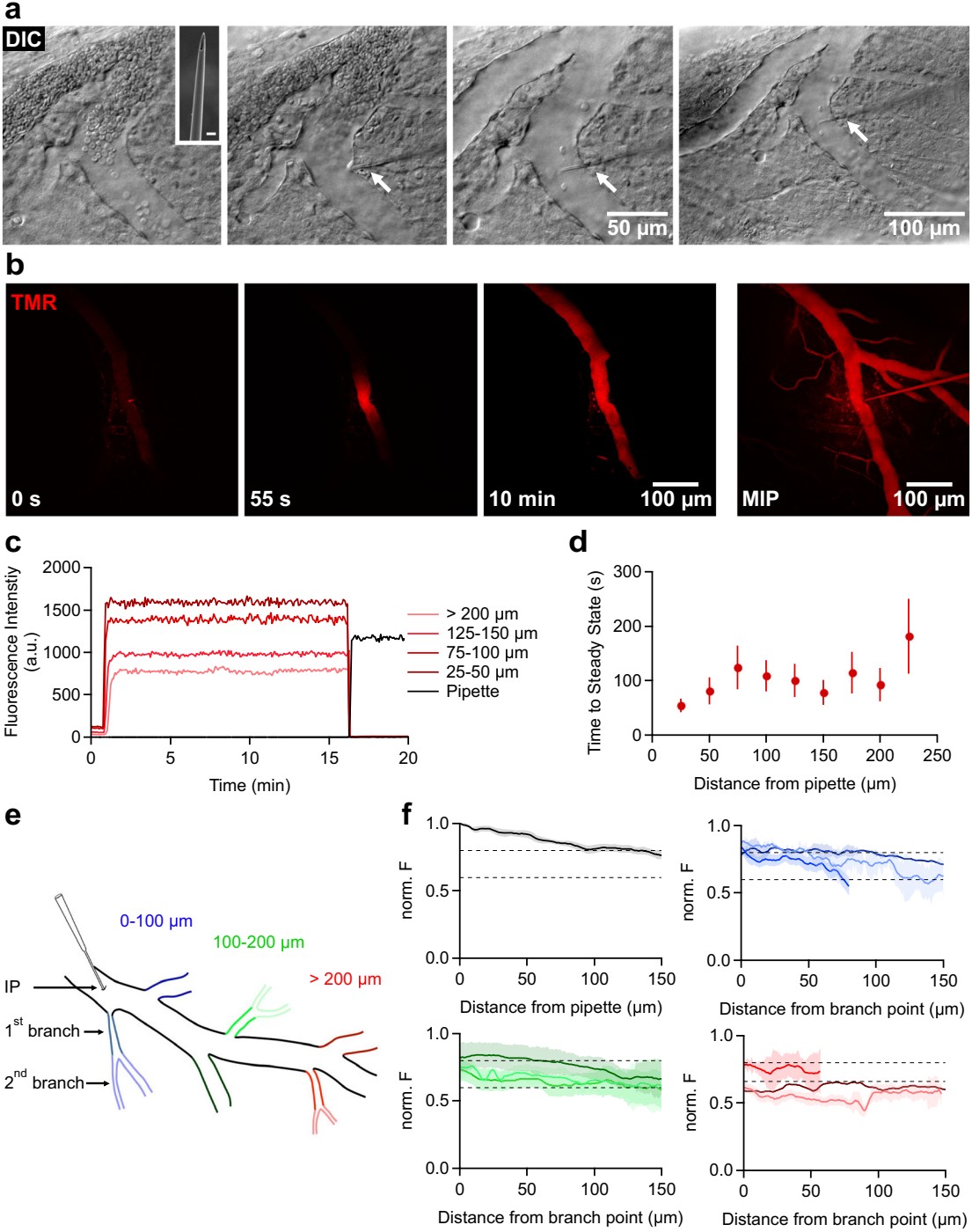

**Fig. 1 | Micro-pipette-based perfusion of capillaries provides control of the composition of the luminal solution and pressure. a** Representative light microscopic (differential interference contrast, DIC) images showing the penetration of a vessel with a glass capillary (4 experiments). Arrow points toward the tip of the pipette. Inset: raster electron micrograph of the beveled tip (scale bar 5 μm). Also see Suppl. movie 1 illustrating the flushing of the vascular lumen. **b** 2P-scans representative of at least 15 experiments. MIP; maximum intensity projection. **c** Intravascular fluorescence. Numbers denote the distance of the measurement from the point of impalement. **d** Luminal steady-state tracer concentration, as assessed by fluorescence measurements, are reached within ~100 s. Data obtained from 15 injections/9 animals; number of assessed ROIs: $n_{IP} = 15$; $n_{0-25\mu m} = 15$;

$n_{25-50\mu m} = 14$; $n_{50-75\mu m} = 12$; $n_{75-100\mu m} = 10$; $n_{10-125\mu m} = 8$; $n_{125-150\mu m} = 7$; $n_{150-175\mu m} = 5$; $n_{175-200\mu m} = 5$; $n_{>200\mu m} = 5$. Data are presented as mean values +/- SEM. **e** Schematic classification of analyzed compartments of the vascular tree for assessing intravascular fluorescence intensities. Intensities measured within ROIs in the compartments and are shown in (f) (11 injections/6 animals). **f** Normalized intravascular fluorescence intensity profiles measured along the injected blood vessels (black) and vessels branching off at 0–100 μm (blue), 100–200 μm (green) and >200 μm (red) from the injection point. Dark lines represent the moving average of 5 measurements, shaded regions represent SEM. Data obtained from 11 injections/5 animals. Source data to panels of Fig. 1 are provided as a Source Data file.

It is now understood that the BBB is not just a constant filter for specific molecules entering and leaving the brain parenchyma but that transport is regulated in a very dynamic manner, depending on the biological states of the body: it is bidirectionally linked to neuronal activity, stress, sleep, changes of nutrition and it is coupled to the immune system[3]. Therefore, it is not surprising that disease-associated changes of the BBB are receiving increasingly more attention and causal roles of BBB alterations have, for example, been suggested for Alzheimer's disease and epilepsy[8,9]. Coupling of transport and BBB permeability to body states and systems-level functions is likely based on blood-borne signaling[10].

While significant progress on this topic has been made in recent years[11–15], a number of major questions regarding this dynamic regulation of the BBB and the specific facilitation of the passage of some molecules versus others are still mainly unresolved: a) What are the key molecules and pathways essential for the dynamic and pathological alterations of the BBB?, b) how and when are immune cells recruited across the BBB, c) do ECs and BBB properties differ between brain regions and across the smaller vessels between arterioles, venules and capillaries, d) which pattern of neuronal activity influences the transport and permeability of the BBB, e) conversely, how does the regulated transport of metabolites and neuroactive molecules at the BBB influence the activity of neurons in health and disease?

A more detailed understanding of BBB properties is not only fundamental to biology but has also great translational potential for improving drug delivery to the brain. Many CNS-active molecules fail to reach the brain parenchyma following oral or intravenous administration as they do not pass or are extruded at the BBB[16]. Moreover, pathophysiological changes can strongly impact drug transport into the brain. Therefore, defining modes of hindrance and extrusion at the BBB and identifying ways to modulate them for improved drug delivery holds great promise to advance medical treatment of CNS diseases[17]. Current approaches to improve CNS drug delivery successfully concentrated on altering EC properties and capitalized on their transcytotic pathways to overcome the BBB[18–20] rather than risking a general leakage by weakening the ECs' intercellular tight junctions or a metabolic imbalance by blocking BBB transporters. Validation of these strategies in the human brain vasculature would be desirable, as well as more detailed understanding on the regulation of intracellular trafficking towards transcytosis, recycling back to the bloodstream or degradation[7]. Further, it has recently been discovered that in inflammatory diseases where the endothelial BBB is disturbed astrocytes respond and re-build a second barrier by expressing tight junctions[21] potentially complicating drug delivery in the disease condition.

With the currently available methodology, answering these open questions is rather difficult. Approaches are required that allow analysis of the native BBB, which has been in interaction with the healthy or the diseased brain and was under the control of systemically circulating signaling molecules. Furthermore, experimentally controlling the intra-luminal solution is of paramount importance in order to apply or interfere with BBB-activating molecules, cells, drug candidates or fluorescent tracers. Suitable experimental approaches should be applicable to various brain regions in different species, to capillaries, venules and arterioles and also to other organs for comparative studies of the BBB. Addressing the interdependence of the BBB and neuronal activity requires combining BBB analysis with recording and stimulating neuronal activity[7]. Ideally, experimental approaches should allow the visualization of cellular and sub-cellular details of the NVU in situ with high spatial and temporal resolution to study the barrier-forming intercellular interfaces of ECs as well as of BBB-penetrating immune cells and to optically track BBB permeation of labelled molecules[22]. In particular, it will be important to visualize the cytoplasm of ECs to analyze the different transcytotic pathways,

as the regulation of transcytosis[23] is emerging as a critical mechanism of maintaining BBB integrity and facilitating drug targeting[24].

Many useful and sophisticated in vitro models of the BBB based on cultured cells have been developed and successfully used to address specific questions[25]. However, as BBB properties are not intrinsic to ECs but arise from an interaction with both the brain and the rest of the body via circulating factors or cells, it is clear that such models, while certainly very useful for specific questions, are limited in their ability to replicate many aspects of the healthy and diseased BBB.

In vivo approaches to study the BBB with high-resolution imaging on the other hand provide a gold standard in looking at the healthy or diseased native BBB in its physiological environment[26]. However, not all brain regions and developmental stages are amenable to intravital microscopy and optical resolution at subcellular levels is difficult to achieve[27]. The bloodstream can be supplemented with drugs, tracers and signaling molecules in vivo by intravenous injection but it is usually difficult to maintain their intraluminal concentration at a constant level due to systemic metabolism and excretion[28]. Furthermore, if experimental substances are applied systemically, it is hard to precisely define how, where and when they entered the brain parenchyma[29]. Finally, the possibilities to systematically alter the composition of the intraluminal and brain interstitial solutions is limited, as blood flow and oxygen content are crucial for brain vitality during observation and the extracellular solution of the brain parenchyma is not easily accessible, respectively.

Taken together, for the currently available approaches to study the BBB there is a general trade-off between experimental controllability and accessibility on the one hand and biological validity on the other, hampering the ability to answer fundamental biological and pressing translational questions.

Here, we report an approach to study the native BBB in situ, which unites systematic accessibility with high validity and flexibility. This approach is based on combining micro-perfusion of capillaries in acute brain slices with high-resolution multiphoton microscopy. It allows to systematically study BBB transport, permeation of cells and molecules in real-time, resolves subcellular details of cells of the NVU and offers control over luminal solution stream and pressure. We furthermore showed that this approach can be combined with cellular electrophysiology to record and evoke neuronal activity and allowed comparison of drug activity following luminal versus parenchymal application. We validated and applied this approach to human and murine brain under healthy, diseased and damaged conditions.

## Results

To systematically and quantitatively study regional properties and locally restricted mechanisms of the native BBB, we aimed at controlling the composition of the solutions on both sides of the barrier in acutely prepared mouse cortical brain slices. While the parenchymal side of the BBB is easily and routinely controlled in a brain slice recording chamber via the perfusion medium, the lumen of the vascular system remained experimentally inaccessible and capillaries collapse in the absence of the blood or a solution stream. We achieved access to the luminal side of the BBB by inserting glass capillaries into vessels which enabled us to re-establish a solution stream in the vascular system of a brain slice and gave us full control of the composition of the luminal solution. To this end, we beveled injection pipettes to a ~3 μm diameter and a sharp flat-angled tip (Fig. 1a). Under microscopic control (Fig. 1a, b), dye-filled pipettes (100 μM biocytin-tetramethylrhodamine, TMR) were carefully inserted into an acute brain slice (300 μm) and moved towards the border of a vessel with the help of micro-manipulators (for images of the setup see Suppl. Figure 1a–d). The type of vessels approached (artery, arteriole, capillary, venule, vein) could clearly be discerned in the DIC or Dodt image, based on

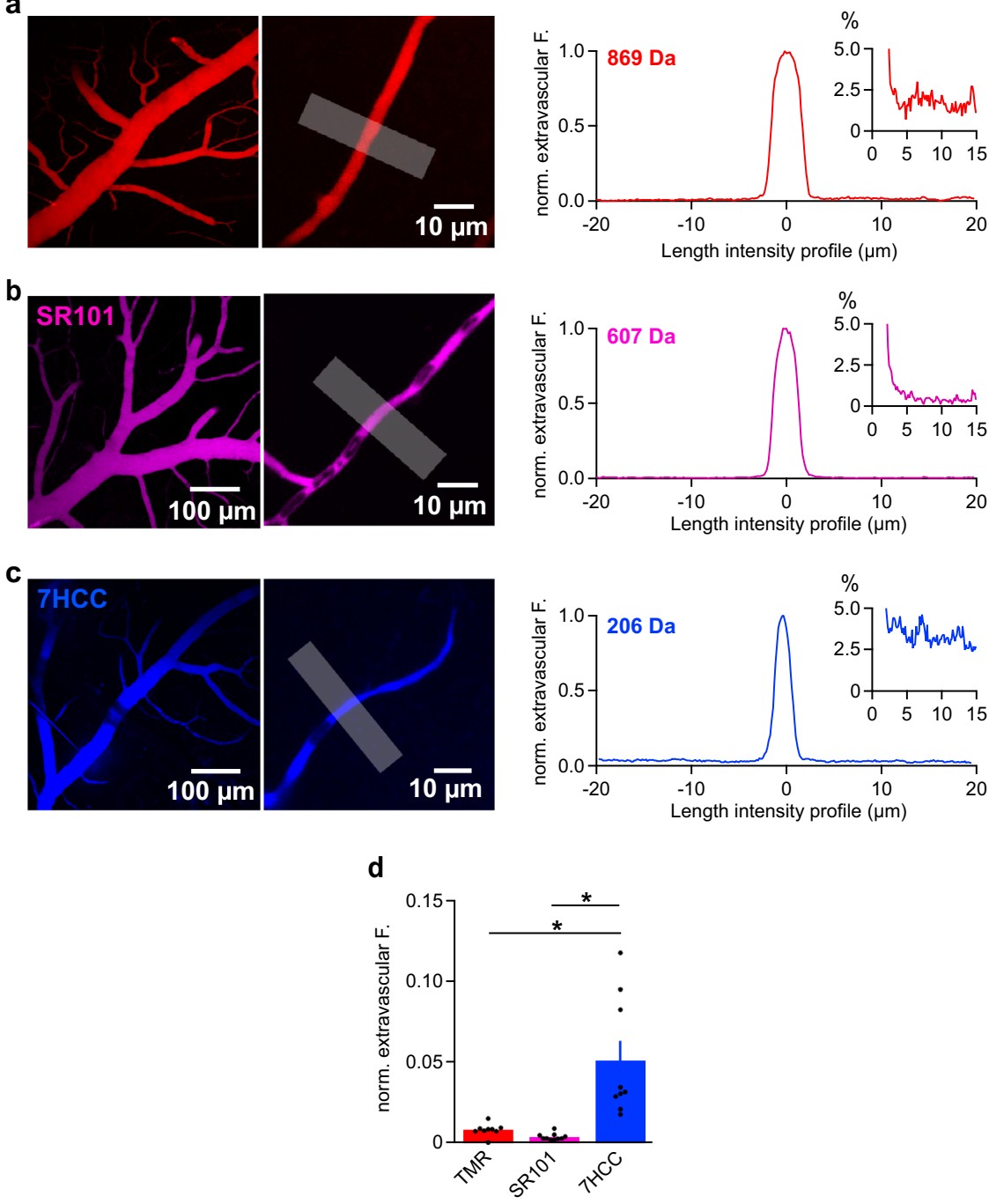

**Fig. 2 | Tracers of variable molecular size show differential diffusion rates across the BBB. a** MIP of TMR-injected vascular tree and a single frame of a capillary (left) and the spatial fluorescence profile of an ROI (white line) across a capillary at 30 min (left) with less than 2% extravascular fluorescence (see inset). Images represent 9 experiments. **b** MIP at $t_{30}$ of SR101 injection (left) and the line profile of an ROI across a capillary (right) showing an extravascular fluorescence below 1% (see inset). Images are representative of 5 experiments. **c** MIP of the vascular tree and a single frame of a capillary recorded at $t_{30}$ of 7HCC injection (left) and the spatial fluorescence profile of an ROI (white line) across the capillary at 30 min (right) showing 3–4% extravascular fluorescence (see inset). Images are representative of 5 experiments. **d** Comparison of fluorescence intensities measured within ROIs placed on single frames of capillaries at 30 min after normalization to the intravascular fluorescence. Data are graphed as mean ± SEM from 19 injections/8 animals, ROIs: $n_{TMR} = 9$, $n_{SR101} = 9$, $n_{7HCC} = 10$. Statistical significance was calculated using a one-way ANOVA with Tukey's post *hoc* test (*, $P < 0.05$). The $P$ values between TMR/7HCC and SR101/7HCC were 0.0006 and 0.0001, respectively. Source data to panels of Fig. 2 are provided as a Source Data file.

their diameter and thickness of the wall[30]. In general, all types of vessels could be penetrated when approached from the side by the tip of the injection pipette, except capillaries. Capillaries were too elastic and could be pushed for several hundred μm through the brain slice without rupture and without penetration by the pipette tip. Arterioles could be penetrated but the success rate was low (<20%) as often the tip of the glass pipette broke at or in the wall of the vessel. Therefore, we chose venules (diameter 20–50 μm) which yielded a high success rate (~80–90%) for experiments. The piercing was done at a slice depth of 30–50 μm which still permitted a good visualization of the pipette tip with wide-field microscopy (DIC or Dodt contrast) but the dye-filled arborization of the vascular system was followed deep into the brain

slice with two-photon (2 P) microscopy (Fig. 1b). After pushing the tip of the pipette ~5–10 μm into the lumen of the venule, a slight pressure (~100 mmHg) was applied to the back of the pipette. This resulted in an immediate perfusion of the vessel system, thereby flushing away blood cells (Suppl. Movie 1) and filling capillaries and the venous system, causing them to reach their full diameter (approx. +20%, Fig. 1a). The luminal spread of the dye could be observed simultaneously with 2 P scanning microscopy (Fig. 1b). Within a minute, steady-state fluorescence levels were achieved and a multitude of vascular branches became visible (Fig. 1b–d). Near the injection point, the dye concentration in the lumen was comparable to that in the pipette (relative fluorescence 118 ± 7.5%) and then dropped with distance due to vascular branching and the resulting increase in total vascular diameter (Fig. 1e, f). Note that the dye concentrations remained constant in large and small vessels as long as pressure was applied (Fig. 1c). The fluorescence of the TMR-filled vascular system appeared with a high contrast against the brain parenchyma, suggesting that BBB integrity was well maintained in acute slices and that tight junction protein complexes do not allow a polar tracer such as TMR to cross the BBB[31].

We perfused a range of tracer molecules and tested whether they cross the BBB by quantifying the fraction of extravascular fluorescence nearby the vessel (Fig. 2). The extravascular fluorescence of TMR (869 Da) steeply dropped to ~1% of the luminal fluorescence across the vascular wall (Fig. 2a, d). Similar results were obtained for the slightly smaller tracer sulforhodamine 101 (SR101, 607 Da, 400 μM, Fig. 2b, d) indicating that these two tracer molecules are too large to permeate the BBB[32]. In contrast, the small tracer 7-hydroxycoumarin-3-carboxylic acid (7HCC, 206 Da, 500 μM) reached significantly higher levels outside the vessel (~5-fold, Fig. 2c, d) suggesting that it is able to cross the BBB paracellularly.

We next tested whether molecules integrating in the endothelial cell membrane can bypass the tight junctional complex at the interface of neighboring ECs via membrane-delimited diffusion. For this, we perfused the styryl dye FM1-43 (40 μM) into the vascular lumen. FM1-43 becomes fluorescent when it integrates into cell membranes. Integration into membranes is reversible and membranes de-stain within minutes after removal of FM1-43. This probe cannot cross phospholipid bilayers and only partitions into the outer leaflet of the membrane (when applied from outside) while freely diffusing laterally within the outer leaflet (Fig. 3a)[33–36]. FM1-43 strongly stained the luminal walls of smaller and larger vessels (Fig. 3b). Higher magnification of the fluorescent image and superposition onto the Laser-Dodt channel to visualize cells showed that the abluminal membrane of ECs did not stain within the first minutes (Fig. 3c, i vs ii and Suppl. Figure 2a) suggesting that tight junction complexes slow down membrane-delimited diffusion. However, extended live cell imaging demonstrated that after ~30 min the abluminal membrane started to stain and accumulated FM1-43 fluorescence (Fig. 3c, d and Suppl. Figure 2a). When we applied 400 μM FM1-43 (i.e. 10-fold increase), the abluminal walls were stained much earlier and became clearly visible already after 5 min (Fig. 3e, f). We neither observed fluorescent punctae nor stained organelles in ECs as would be expected if FM1-43 reached the abluminal membrane by either transcytosis[23] or intracellular diffusion. In further experiments, we perfused DyLight 488-labelled Lycopersicon esculentum agglutinin (tomato lectin) into the vasculature. Tomato lectin binds to glycoproteins containing complex-type and high mannose-type N-glycans present on the luminal membrane of ECs[37] and produced a similar luminal membrane staining as FM1-43 (Fig. 3g). However, we did not observe labelling of the abluminal membrane of ECs even after 30 min (Fig. 3g). This was expected as, unlike FM1-43, the tomato lectin target cannot undergo free lateral diffusion and is integrated in the luminal endothelial glycocalyx[38]. Thus, the data indicates that intra-membrane diffusion across the junctional complexes linking ECs occurs and that molecules with amphiphilic properties similar to FM1-43 might be utilized as shuttles for drug delivery

to the brain. To assess whether FM1-43 unbinds from the lipids of the abluminal membrane of ECs and can be found in tissue surrounding vessels, we quantified the average FM1-43 fluorescence intensity in regions defined along vessels and normalized it on the brightness of the luminal membrane. For comparison we performed the same analysis following staining with tomato lection which was only found on the luminal membrane. As shown in Suppl Fig. 2b and c, FM1-43 reached significantly higher levels in the brain parenchyma supporting the view that this molecule enters the brain tissue after bypassing junctional complexes by membrane-delimited diffusion. Note that cells selected for the analysis of FM1-43 diffusion to the abluminal membrane showed a flat shape largely contained within the outline of capillaries (Fig. 3c, e, h). This appearance is characteristic of ECs whereas PCs show a rounder and more bulky shape clearly protruding from capillaries (Fig. 4h and Suppl Fig. 2d).

Perfusion of both FM1-43 and tomato lectin revealed a rhombic mesh of fluorescent strings along the luminal walls of ECs which could be most easily identified on larger vessels (Fig. 3h (left) and j). A similar staining pattern was observed upon immunolabelling of adherens junctions with VE-cadherin primary antibody (Fig. h, fixed tissue, right). Thus, these fluorescent strings likely represent regions where FM1-43-stained membranes of neighboring ECs are in contact and parallel to each other and are connected by tight and adherens junction proteins. This supports the view that visualizing these strings by intraluminal FM1-43 application may be used to assess the extent of ECs and trace the path of the line of junctional complexes in between them (Fig. 3i). The area and shape of individual ECs in the BBB has, to the best of our knowledge, not been systematically reported before. In FM1-43-stained vessels, we determined the average surface area of ECs to be 182 ± 26.4 μm² and 384 ± 46.2 μm² in capillaries and venules, respectively (Fig. 3j, k, see methods for details of the calculation). The calculated surface areas using tomato lectin staining were very comparable (153 ± 17.2 μm² and 368 ± 15.6 μm² in capillaries and venules, respectively) (Fig. 3j, k). The luminal cellular surface area is a key parameter as it determines the rate of diffusion of membrane-permeable molecules from the blood into ECs and thereby the extrusional transport load per cell. Knowing this area also permits to estimate transporter densities from cellular transport rates or from quantifying the number of transporters per length of a vessel. When compared to the surface area of the venule, ECs consistently cover ~19% of the unit surface area of capillaries (Fig. 3). This implies that ~8 ECs are required to maintain the BBB along a ~83 μm segment of capillaries. Single ECs are on average ~2-fold larger in venules but occupy a smaller fraction of the vessel's luminal surface area (~5%). The size of ECs in venules grows with the vessel's diameter (and thus surface area) linearly but not proportionally, as observed for capillaries (Fig. 3k).

A number of membrane-permeable molecules, which generally permeate cells, cannot pass the BBB because ECs strongly express efflux mechanisms at their luminal membrane[16] that potently and actively extrude such molecules back into the lumen of the vessel (Fig. 4). The ATP-binding cassette (ABC) transporters ABCB1/MDR1, ABCG2/BCRP1 and ABCC1/MRP-1 are considered the most prominent transporters in ECs of the BBB[39]. It has so far remained difficult to quantitatively study the regional and cellular potency of these efflux pumps in ECs of the native BBB. Here, we developed three assays to study the functionality of ABC transporters in live ECs in the combined brain slice and capillary perfusion model (Fig. 4).

For the first assay, we intraluminally perfused the membrane-permeable fluorescent dye rhodamine123 (15 μM), which potently stains mitochondria when applied at nanomolar concentrations[40–42]. However, under control conditions rhodamine123 did not label mitochondria in ECs even after perfusion of the vasculature for 30 min with 15 μM (Fig. 4a). This suggested that ECs possess remarkably potent rhodamine123 extrusion mechanisms, which keep

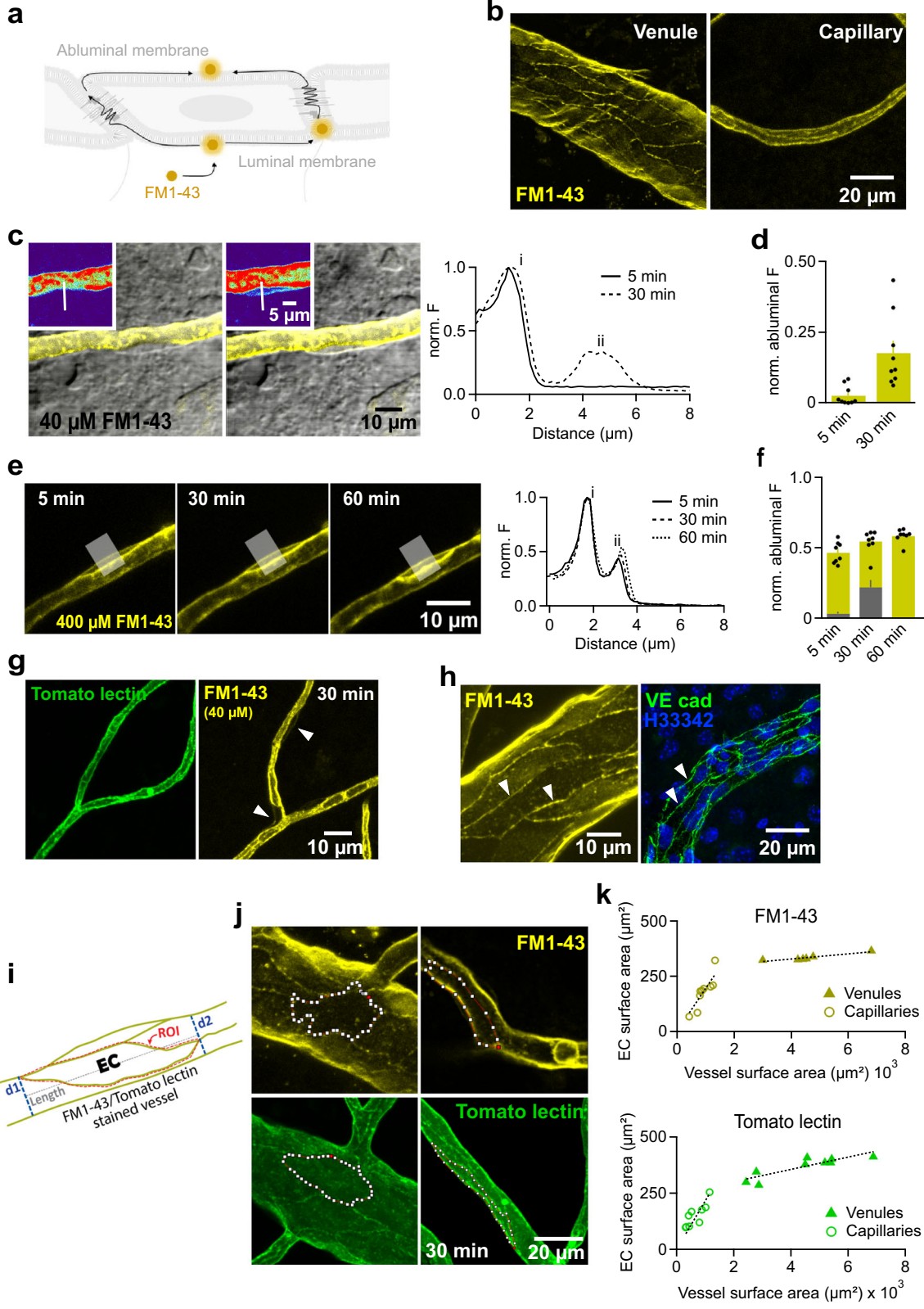

the intra-endothelial cell concentration well below the nanomolar range despite continued application of the dye such that it did not accumulate in mitochondria. Rhodamine123 is a substrate of the ABCB1[43], ABCC1[44] and ABCG2[45] transporters. ABCB1 and ABCC1[46] can be inhibited by verapamil[47]. Bath application of verapamil (200 μM, pre-applied for 15 min) drastically changed the experimental outcome in this assay. Already early during the experiment, a patchy

staining of ECs along both capillaries and venules was clearly visible, which continuously increased during the recording time (Fig. 4b, left). The patchy staining of the walls of the vessels likely represented labelled mitochondria in ECs as another mitochondrial stain (Mito-Tracker Orange CMTMRos 50 μM) produced quantitatively and qualitatively a very similar labelling pattern under the same conditions (Fig. 4b, right).

**Fig. 3 | Membrane dyes outline endothelial cells and can be used to measure their surface area across the vascular tree. a** Schematic representation of how FM1-43 incorporates into the outer leaflet of the luminal and abluminal membranes. **b** MIPs of a FM1-43 perfused venule (left) and capillary (right) at 30 min. **c**, FM1-43 fluorescence (yellow and false color, inset) in the luminal and abluminal membranes of a capillary EC. FM1-43 fluorescence is superimposed on a laser Dodt scan. Right panel shows the spatial fluorescence profile at 5 and 30 min. **d** Summary of fluorescence intensity of the abluminal membrane normalized to the luminal membrane ($n_{ROI}$=10). **e** A 400 μM FM1-43 injection outlines an EC in a capillary at a faster rate. The right panel shows the spatial fluorescence profiles at the times indicated of an line ROI (white line). **f** Quantified fluorescence intensity in the abluminal membrane ($n_{ROI}$ = 8). **g** Tomato lectin outlines ECs across capillaries at

30 min in a pattern similar to that obtained with the FM1-43 probe (in 40 μM concentration). **h** MIPs of a FM1-43-injected blood venule showing EC borders (arrowheads)(left). Images are representative for 10 injections in slices of 4 animals. Right, immunofluorescent labelling of adherens junctions (VE-cad) outlining the ECs (arrowheads) (Hoechst 33342 counterstain). The image is representative of 6 slices/3 animals. **i** Cartoon illustrating how the surface area of EC was computed based on the outline of an EC (region of interest, ROI). **j** Outlined ECs in FM1-43 (top) and tomato lectin (bottom) -labelled venules (left) and capillaries (right) can morphometrically be analyzed (see methods for details). **k** The surface area of ECs plotted versus the surface area of the vessels (top panel, slope 0.195) and tomato lectin (bottom panel, slope 0.216). Source data to panels of Fig. 3 are provided as a Source Data file. Data are presented as mean values +/- SEM.

Secondly, we assayed the transport activity of ECs using calcein-AM. Calcein-AM is membrane permeable and non-fluorescent until the AM group is cleaved off by intracellular esterases, releasing fluorescent calcein that cannot pass membranes anymore. Calcein-AM is a known substrate of ABCB1 and ABCC1, while fluorescent calcein is a substrate of only ABCC1[48]. Similar to the rhodamine123 experiments, intraluminal application of calcein-AM (20 μM) in the absence of ABC transporter inhibitors did not lead to accumulation of green fluorescence in ECs (Fig. 4c). In contrast, when blocking ABCB1/ABCG2[49] and ABCC1 with Elacridar (1 μM[50]) and Probenecid (1 mM[51]), respectively, ECs strongly accumulated calcein along capillaries and venules (Fig. 4c, d). Application of Elacridar alone, but not of probenecid, was sufficient to induce staining of ECs (Fig. 4e) suggesting that ABCB1 (calcein-AM is not transported by ABCG2[52]) is the primary transporters extruding calcein-AM from ECs. In contrast, the transport rate of ABCC1 alone was not sufficient to prevent accumulation of calcein in ECs.

Not only ECs but also PCs are found along vessels. To assess whether some of the cells which accumulate calcein may represent PCs, we labelled PCs with a PC-specific live cell tracer, neurotrace 500/525[53] (1:250), and perfused a red-shifted calcein-AM version (20 μM) in the presence of Elacridar (1 μM) and probenecid (0.6 mM) to induce calcein accumulation in ECs. As shown in Fig. 4f, the red-shifted calcein selectively accumulated in ECs and not in PCs.

The third assay was based on the membrane-permeable DNA stain Hoechst33342 (Hoechst) that has a picomolar affinity for double-stranded DNA[54]. Hoechst was bath-applied at high concentrations (5-10 mM) and combined with intraluminal FM1-43 application to visualize the outline of ECs. However, while Hoechst strongly stained nuclei of neurons and glial cells it never stained ECs (Suppl. Figure 3a), demonstrating that ECs can generate and maintain very high and very local dye concentrations across their membranes. Only after we subjected slices to chemical fixation, which inactivates all transporters, were the nuclei of ECs and other cells clearly stained by Hoechst (Suppl Fig. 3b).

We further explored the suitability of our approach of micro-perfusion of capillaries to detect, track and quantify acute chemical or physical damage of the BBB in real-time. For this purpose, we introduced a spatially restricted and acute physical lesion of a capillary branch by exposing its wall to strong laser illumination (see methods). While normally TMR is highly restricted to the lumen, after the lesion we observed immediate and profound extravasation of the dye. The parenchymal fluorescence strongly increased within seconds and remained elevated for the duration of the experiment (Fig. 5a). This increase in extra-vascular parenchymal fluorescence was confined to the site of the laser damage while other capillaries were unaffected (Fig. 5a).

DMSO induces a concentration-dependent chemical lesion of the BBB such that intravenously injected proteins (>40 kDa) can pass through the BBB and were found in the extracellular space of the brain parenchyma[55]. While DMSO treatment has been experimentally considered as a potential way to improve CNS drug delivery[56,57], the mechanism of its action is still not fully resolved. DMSO-induced

damage is thought to be caused either by enhanced transcytosis or altered integrity of tight junctions[55]. To observe the action of DMSO during live-cell high resolution imaging, we micro-perfused fluorescently-labelled bovine serum albumin (BSA-Alexa488, 66.5 kDa) into the vasculature. Under control conditions, BSA-Alexa488 was only detected within the vascular lumen (Fig. 5b). In contrast, when 10% DMSO was co-micro-perfused with BSA-Alexa488, we detected clustered droplets of BSA-Alexa488 in the CNS parenchyma in the vicinity of smaller and larger vessels indicating that the protein had crossed the BBB (Fig. 5b, c). However, we never found ECs labelled with BSA-Alexa488 and their cytoplasm remained invisible and free of fluorescent vesicles, suggesting that BSA may have reached the parenchyma by passing through DMSO-weakened tight junctions rather than by increased transcytosis via ECs. The droplets showed little mobility and progressively accumulated over time indicating that they could have been phagocytosed by resident cells. Immunohistochemical staining of microglial cells (TMEM119[58]) did not reveal a co-localization of BSA-Alexa488 and showed unaltered microglial cell densities between the control and DMSO condition (Fig. 5d). In contrast, labelling for the F4/80 antigen (putative macrophages[59,60], see for discussion[59]), which normally is not found in the healthy brain (Fig. 5e), revealed a clear co-localization with BSA-Alexa488 (Fig. 5d) and showed that the DMSO treatment potently induced F4/80-positive cells within 30 min (Fig. 5e). The morphology of F4/80 labelled cells closely resembled that of perivascular macrophages which are known to phagocytose material of the size of BSA[60-62]. These myeloid cells have been reported to normally reside in the perivascular space of venules and arterioles, phagocytose material of 10–70 kDa and play multiple roles in cerebral diseases[60]. In time-lapse imaging, we did not observe diffuse leakage of BSA-Alexa488 into the brain parenchyma under DMSO application but rather a progressive appearance of clusters of fluorescent droplets. This observation is most consistent with a scenario in which putative macrophages in the perivascular space effectively phagocytosed BSA-Alexa488 material that diffused across DMSO-injured tight junctions and thereby prevented a diffusive appearance of the labelled proteins in the extracellular space. Weakening of tight junctions by DMSO is also suggested by our observation of a small but consistent spatial leakage gradient of TMR into the parenchyma (Fig. 5e, f). This leakage is much smaller than the one observed after laser damage but clearly exceeds levels around untreated vessels under control conditions (Fig. 5g). Our data also showed that at least for a time frame of 30–60 min, BSA-Alexa488 is not phagocytosed by TMEM119-positive microglial cells (Fig. 5d, e).

The DMSO experiments demonstrate damage to the BBB which might be explained by a partial opening of tight junctions but they do not directly reveal the mechanism causing this damage. To lend support to the view that DMSO affects the tightness of junctional protein complexes, we perfused a hyperosmolar mannitol solution (15%) for comparison. Hyperosmolar mannitol is known to temporarily open tight junctions by a combination of vasodilation and shrinkage of ECs[63,64]. Within 10 min, mannitol induced a similar progressive appearance of clusters of BSA-Alexa488 along vessels

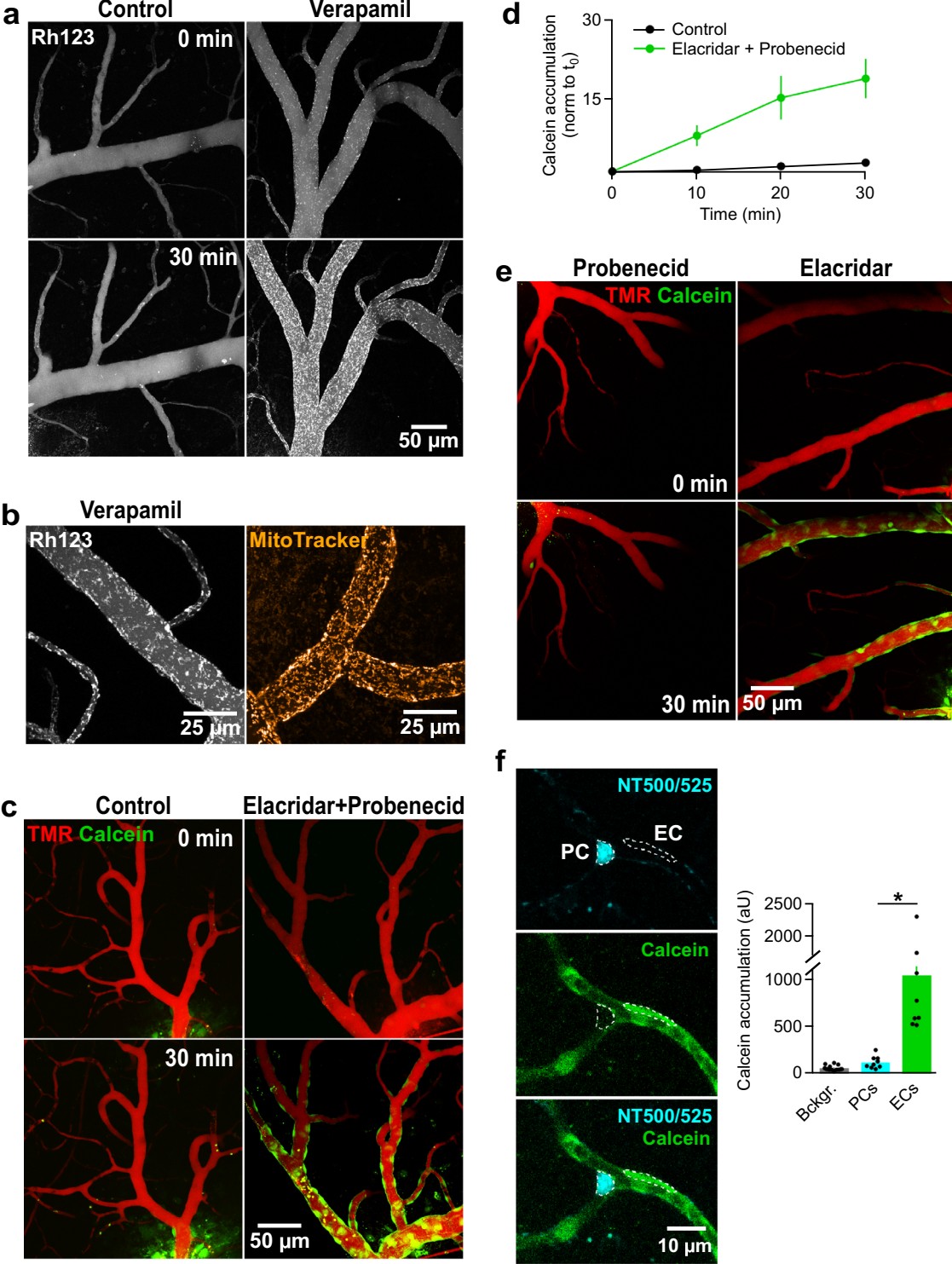

(Fig. 6a). After 30 min, the clusters became dense and clearly marked the cells they were contained in. Cells containing BSA-Alexa488 were most frequently found along venules when compared to capillaries (Fig. 6b). As observed following DMSO application, cells which had taken up BSA-Alexa488 were stained by anti-F4/80 antibodies (Fig. 6c) suggesting the same origin. When we co-applied TMR and mannitol, we found a similar small but consistent leakage across the BBB and extravasation of this tracer (Fig. 6d, e). To assess the osmotic effect of mannitol, we tracked the diameter of blood vessels upon perfusing mannitol. The diameter of capillaries significantly increased after 30 min mannitol perfusion by $8.4 \pm 2.5\%$ compared to

the diameter at the beginning of the experiment (after 5 min perfusion, $n = 8$). In the absence of mannitol, the diameter remained unchanged upon perfusing TMR solely ($1.3 \pm 2.2\%$, $n = 7$). The diameter of venules did not change significantly under the same conditions ($1.9 \pm 0.4$, $n = 10$, and $2.0 \pm 1.1$, $n = 6$, with and without mannitol, respectively). Taken together, hyperosmolar mannitol closely reproduced the findings detected after perfusion of DMSO suggesting, but not proving, that the two manipulations act via the same mechanism and partially open tight junctions.

Breakdown of the BBB in epileptic patients and animal models of epilepsy has been previously reported but the regionality and degree

**Fig. 4 | Assessing cellular transport rates of endothelial efflux systems. a** MIPs recorded at $t_0$ (top) and $t_{30}$ (bottom) under control conditions (left) and ABCB1 inhibition with verapamil (right). Accumulation of rhodamine123 (Rh123) in mitochondria of endothelial cells is only visible when verapamil is applied. Rhodamine123 is hardly seen in cells around the vessel likely because the endothelial cell cytoplasm already contains a low Rhodamine123 concentration due to remaining extrusion activity and there is a strong dilution of the dye in the surrounding 3D environment. Images are representative of 11 experiments. **b** MIPs acquired at $t_{30}$ after rhodamine123 and MitoTracker injections. Note the very similar staining pattern of both dyes. Images are representative of 5 experiments. **c** MIPs recorded at $t_0$ (top) and $t_{30}$ (bottom) under control conditions (left) and simultaneous efflux protein inhibition of ABCB1/ABCC1 with 1 μM Elacridar and 1 mM probenecid (right). Only in the presence of ABC-transport inhibition did calcein accumulate in endothelial cells. Images are representative of 10 experiments. **d** Quantification of the endothelial fluorescent signal over time. Plot represents the mean normalized fluorescence measured in capillary endothelial cells ($n_{ROI}$=19, 9 injections/9 animals). Data are presented as mean values +/- SEM. **e** MIPs recorded at $t_0$ (top) and $t_{30}$ (bottom) under individual ABCC1 (left) or ABCB1 (right) inhibition with probenecid and Elacridar, respectively. Images are representative of 6 experiments. **f**, Quantification of calcein accumulation in pericytes (PC) at 30 min after calcein-red-orange injection under protein inhibition of ABCB1/ABCC1 with 1 μM Elacridar and 0.6 mM probenecid. PC were prelabeled with Neurotrace 500/525 (NT500/525) (cyan). The recorded individual channels show little to no overlap between calcein and NT500/525 within the PC. The graph shows quantification of the mean fluorescence intensity of the calcein signal within EC ($n_{ROI}$ = 9) and PC ($n_{ROI}$ = 9) in the calcein detection channel compared to the background (Bckgr.) signal ($n_{ROI}$ = 18) (± SEM, 7 injections/3 animals). Statistical significance was calculated using a two-tailed Student's $t$ test (*; $P$ = 0.0004). Source data to panels of Fig. 4 are provided as a Source Data file.

of BBB damage could not be precisely determined[65,66]. To address this issue, we employed the pilocarpine-induced status epilepticus (SE) model[67,68], in which SE initiates a process termed epileptogenesis that finally leads to chronic recurrent seizures and temporal lobe epilepsy[69]. In this animal model of epilepsy, BBB breakdown has been assessed histologically following systemic in vivo administration of tracers[70]. Mice were taken into the experiment 2 h following pilocarpine induced SE (Fig. 7a) and tested for tightness of junctions and leakage across the BBB by micro-injection of the tracer TMR. After 14 injections (n = 3 mice), a small leakage similar to that induced by DMSO or mannitol treatment (Fig. 7b, c) was consistently observed indicating that the integrity of tight junctions was similarly affected after pilocarpine-induced SE in mice.

As it was reported previously that BSA after systemic injection is found in the mouse brain following SE, we also micro-perfused BSA-Alexa488 2 and 24 h after the status (3 and 4 mice, respectively). However, we did not observe extravasation of BSA-Alexa488 resembling that seen under DMSO or mannitol treatment in any of the mice (Fig. 7d).

We further evaluated BBB properties of human epileptic hippocampi (7 patients). Hippocampi were removed for surgical control of medically intractable chronic temporal lobe epilepsy (TLE)[71] and transferred from the operating room to the laboratory in cooled ACSF within 10–15 min. Once in the lab, human tissue underwent identical slicing and experimental procedures as murine tissue. Micro-perfusion was well applicable to human tissue albeit the walls of human vessels appeared sturdier and turned out to be more difficult to penetrate than mouse vessels. Similar to the mouse tissue, BSA-Alexa488 fluorescence was confined to the human vasculature and we did not detect fluorescent droplet-loaded macrophages suggesting the protein does not reach the brain parenchyma (Fig. 7e, f). These data show that the approach of vascular micro-perfusion in brain slices is well applicable to diseased tissue across species and suggest that a single SE in the murine model can cause a lasting leakage for small molecules (TMR) across the BBB. However, the BBB is sufficiently intact to avoid detectable BSA extravasation in the mouse model and human chronic epilepsy.

Previous work reported up-regulated expression of ABCB1 at the BBB post SE[72]. Therefore, we assessed the functionality of ABCB1/ABCC1 transporters in our murine model 2 h post SE. In both sham- and pilocarpine treated mice, perfusing calcein-AM in the vasculature did not result in any noticeable calcein accumulation in ECs suggesting that in both groups there is no relevant reduction in calcein-AM transport rate compared to control mice reported above (Fig. 7g, left column). To test for increased calcein-AM transport rates we partially blocked ABC transporters (0.6 μM Elacridar, 0.6 mM probenecid) to induce a small degree of accumulation of calcein in ECs in the beginning of the experiment (at $t_0$, Fig. 7g, middle column). In ECs of sham-injected mice, the fluorescent levels of calcein increased to more than 4-fold over 30 min (Fig. 7h). In contrast, ECs in slices prepared from pilocarpine-injected mice accumulated only little additional fluorescence over the same period (Fig. 7h), pointing to a substantial increase in the functional activity of the transporters extruding calcein-AM from ECs (ABCB1/ABCC1).

We assessed whether the protein levels of ABC transporters were altered after SE by performing a proteomic analysis of hippocampal tissue from sham- and pilocarpine-injected mice. A number of transporters were detected in samples obtained 4 and 24 h post SE including ABCB1(MDR1) and ABCG2(BCRP1) (Fig. 7i). ABCC1(MRP1), which appeared to play a minor role in calcein-AM extrusion, was not detected. We only measured small changes in ABC-transporter protein levels and counts in post-SE mice fell within ±12% of sham mice (Fig. 7i). Only at 24 h post-SE, some of these alterations reached the level of statistical significance. As the relative changes in the ABC protein levels were small, the proteomic data suggest that the observed 3-4-fold-increase in calcein-AM extrusion rate of ECs from pilocarpine-treated mice can hardly be explained by increased protein expression but rather is due to a functional regulation such a dynamic redistribution of ABC proteins from an intracellular pool to the cell membrane[73].

So far in this study, we have optically monitored interactions at the BBB but in some instances, it might be desirable to monitor a functional response, for example neuronal activity, to assess whether a micro-injected molecule crosses the BBB to a sufficient degree. Measuring a functional response allows to determine whether the drug/signaling molecule reaches the target receptor and if it does so at the relevant concentration. Furthermore, as the brain slice system allows to apply a drug at a defined concentration, both directly to the parenchyma ("bath application") and to the vascular system ("micro-perfusion"), an equivalence dose-ratio at the BBB can be calculated at which both routes of application reach the same effect. We demonstrate this approach by studying BBB permeation of caffeine and its antagonistic effect on presynaptic adenosine receptors (A1)[74]. We monitored synaptic transmission with field potential (fEPSPs) recordings of Schaffer collateral-evoked synaptic responses while perfusing TMR in the vasculature to assess the injection quality and follow the vascular tree (Fig. 8d). These synapses express inhibitory A1 receptors and we first applied the A1-receptor agonist $N^6$-cyclopentyladenosine (CPA, 15 nM) to activate A1 receptors and observed the expected reduction in fEPSPs (Fig. 8a). Bath application of 300 μM caffeine in the continued presence of CPA antagonized the activation of A1 receptors and increased fEPSPs responses (Fig. 8a). In a separate experiment we also pre-applied CPA to activate A1 receptors but additionally applied caffeine by micro-perfusion of capillaries at a later time point (Fig. 8b). Consistent with its BBB permeation, caffeine also antagonized the inhibitory action of CPA and increased fEPSP within ~15 min. The extent of antagonisms was slightly (but not significantly) lower than when bath-applied (Fig. 8c) despite the fact that we injected 30 mM caffeine, suggesting an equivalence dose-ratio of ~100.

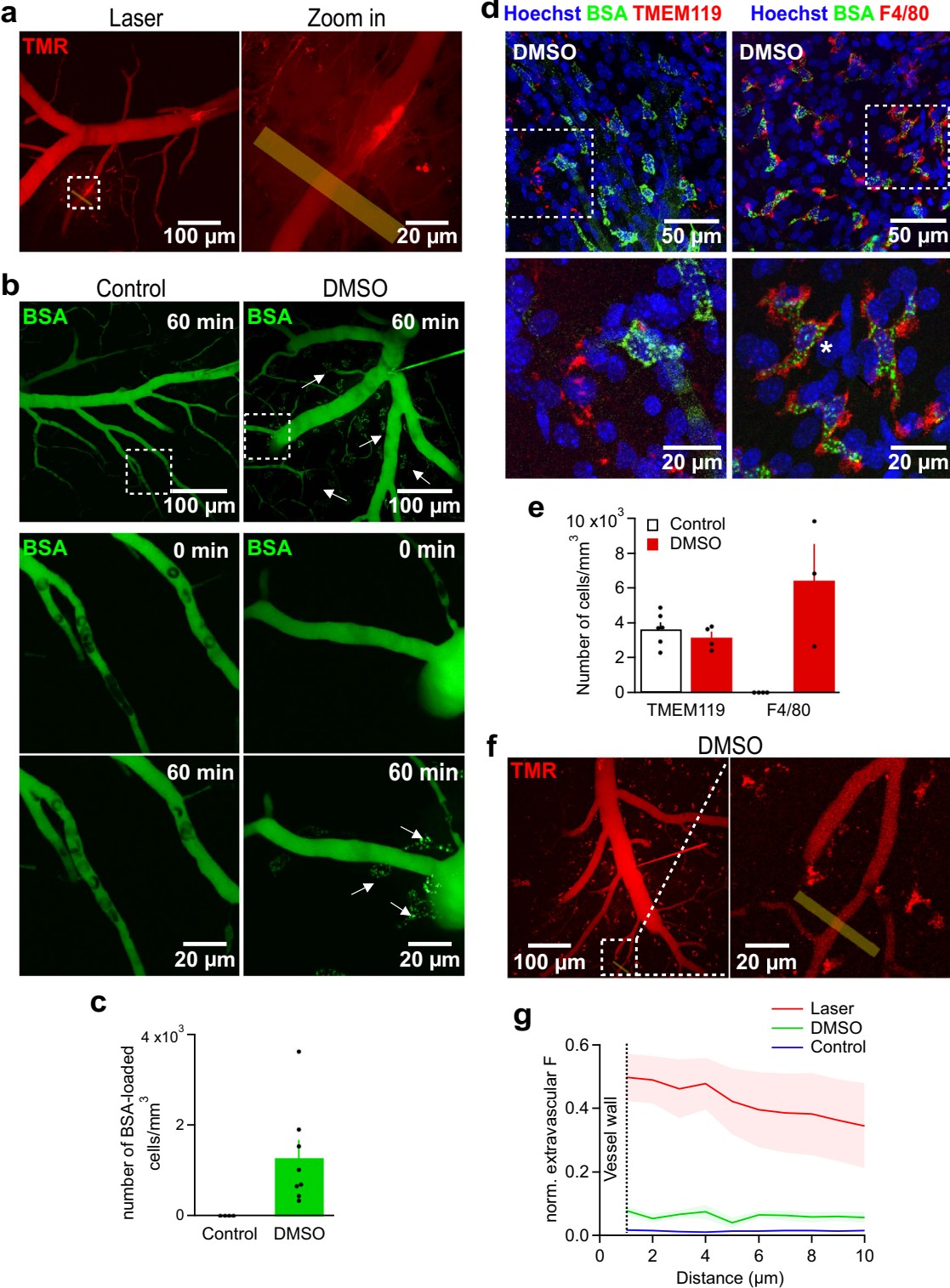

## Discussion

Here, we report the development, characterization and broad applicability of an in situ BBB model based on acute brain slices, microperfusion of capillaries and multi-photon imaging (ISMICAP, in situ micro-perfusion of capillaries). ISMICAP overcomes several limitations of previous models and offers a number of additional advantages. We build on the traditional and long-established technique of preparing acute brain slices for electrophysiological recordings. Decades of research have demonstrated that neurons remain viable for many hours after preparation of the slice. Our results clearly show that the viability of neurons, their electrophysiological properties and the functionality of ABC-transporters functionality are preserved. Calcein-AM, rhodamine and Hoechst were found to be potently extruded from ECs throughout the entire experimental window (up to 8 h following slicing). In fact, when comparing our results to previous assays applied to cultured ECs, it turns out that the activity of ABC transporters is much stronger in ECs in our slices: we did not detect any accumulation of the three ABC transporter substrates mentioned above over a period of at least 60 min and they only became visible after blocking ABC transporters. In contrast, previous studies[75–77] reported that even

**Fig. 5 | Quantifying physical and chemical lesions to the BBB. a** Overview (left) and close-up (right) of a laser-damaged capillary with subsequent tracer diffusion into the surrounding neuronal tissue. **b** MIPs of BSA-Alexa488 (BSA)-injected microvasculature in an overview at $t_{60}$ (top), zoomed-in views of the regions enclosed in dashed squares acquired at $t_0$ (middle) and $t_{60}$ (bottom) under control conditions (left column) and after the addition of DMSO (right column). Arrows indicate extravascular BSA-Alexa488 accumulations. Images are representative of 5 experiments. **c** Quantification of the extravascular BSA-loaded cells potentially referring to macrophages under control conditions (5 injection/5 animals) and after exposure to DMSO (8 injections/6 animals, $n_{ROI} = 8$). **d** Immunofluorescent labelling of microglia (left column, TMEM119) and macrophages (right column, F4/80). White dashed squares indicate zoomed-in regions (below) showing that BSA-Alexa488 only co-localizes with macrophages but not with microglial cells. Micrographs are representative of 5 experiments. **e** The microglial cell density around vessels does not change after DMSO injection, but macrophages, normally absent

from the brain parenchyma, are strongly recruited. Cellular densities for microglia under control conditions (7 injections/3 animals, $n_{ROI} = 6$) and DMSO exposure (4 injections/3 animals, $n_{ROI} = 4$) are compared to macrophage densities under control conditions (7 injections/3 animals) and DMSO exposure (3 injections/2 animals, $n_{ROI} = 3$). **f** DMSO induces a slight leakage of TMR into the parenchyma. The level of leakage is so small that it cannot visually be identified in the image but only became clear upon quantification of fluorescent gradients across the vascular wall (see g). Images are representative of 5 experiments. **g** Fluorescent gradients across the vascular wall normalized to the luminal fluorescence. Note the strong accumulation of dye in the extracellular space of the brain following laser-induced damage of the BBB. TMR accumulation is much weaker after DMSO application when compared to laser-induced damage but clearly stronger than under control conditions. Source data to panels of Fig. 5 are provided as a Source Data file. Data are presented as mean values +/- SEM.

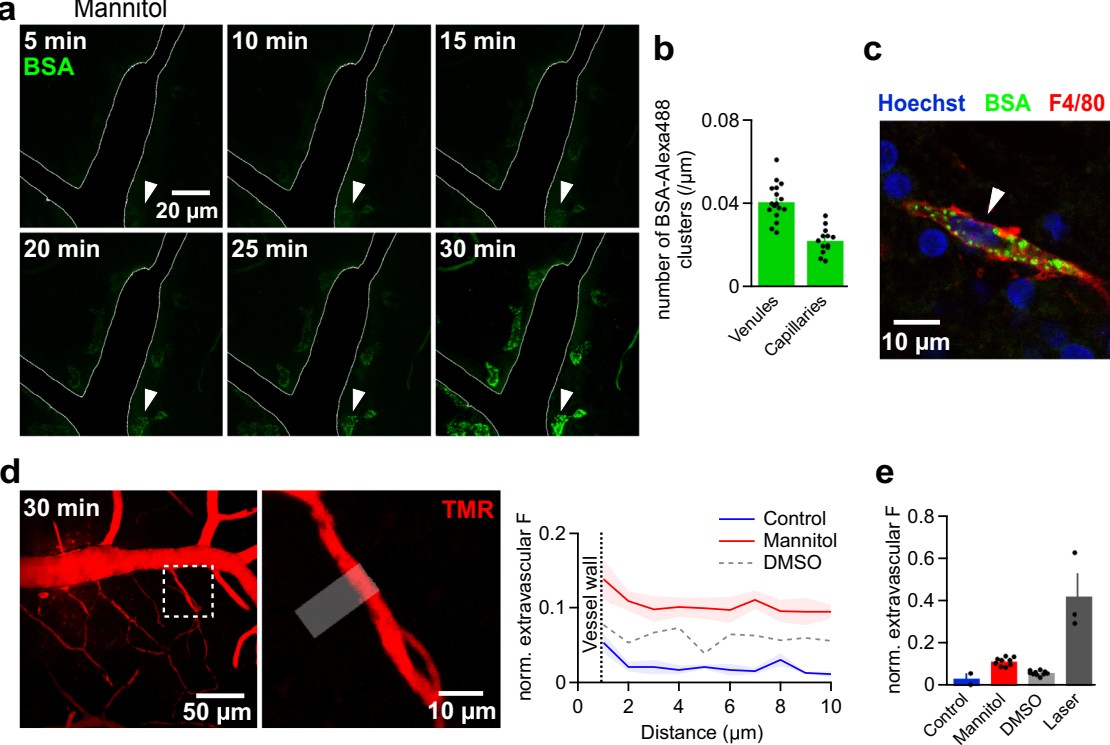

**Fig. 6 | Hyperosmolar mannitol enhances TMR and BSA extravasation to the parenchyma. a** Time lapse images showing the gradual extravasation and subsequent accumulation of BSA-Alexa488 (BSA) in macrophages recruited upon perfusing 15% mannitol into vasculature, which was verified by immunofluorescent labelling (see c). For better clarity of BSA clusters, the intravascular fluorescence was deliberately removed using Fiji software and the vessel was outlined manually. Images are representative of 4 experiments. **b** The number of BSA-Alexa488 clusters within macrophages (per μm length) in the vicinity of venules is twice as high as that next to capillaries (12 injections/ 4 animals, $n_{ROI}$ for venules = 17, $n_{ROI}$ for capillaries = 13). **c** Immunofluorescent labelling of macrophages (F4/80) showing BSA-Alexa488 accumulation within them upon perfusing the vessels with mannitol.

The micrograph is representative of 3 experiments. **d** MIPs of the vascular tree (left) and an enlarged capillary (right) 30 min after co-injection of TMR and 15% mannitol solution. Images are representative of 3 experiments. The graph shows the fluorescence gradients across the vascular wall normalized to the luminal fluorescence (white line) of a capillary compared to that in absence of mannitol (and DMSO, dashed line). **e** Perfusing mannitol for 30 min led to a significantly higher extravascular fluorescence of TMR, normalized to the luminal fluorescence signal ($n_{ROI}$=9, 4 injections/2 animals) at control conditions, yet it was much less than the leakage caused by laser damage. Source data to panels of Fig. 6 are provided as a Source Data file. Data are presented as mean values +/- SEM.

without blockers a clear accumulation of the substrates occurred at a level ~25% of that seen after ABC-transporter inhibition. As ABC transporters are highly ATP-dependent, this means that ECs in our slice preparation can generate ATP for extended periods of time and likely also express more functional transporters than their in vitro counterparts. As a consequence, ISMICAP can significantly reduce the risk of underestimating efflux phenomena during drug discovery research.

Many useful and sophisticated in vitro models of the BBB ideally suited for certain purposes are currently available and they often provide the advantage of applicability for high throughput

analyses[78]. However, an obvious limitation of in vitro models is that they are not composed of native tissue and as a consequence it is hard to achieve and prove the validity of in vitro models in all aspects. In vitro, a BBB is grown with a reduced cellular micro-environment, in the absence of normally circulating factors and without dynamic exchange with the rest of the body. Furthermore, in vitro models lack a solution stream and intravascular pressure, which is important for several physiological responses and the regulation of the NVU[79–81]. Finally, studying the role of the BBB in specific pathological contexts would require to completely

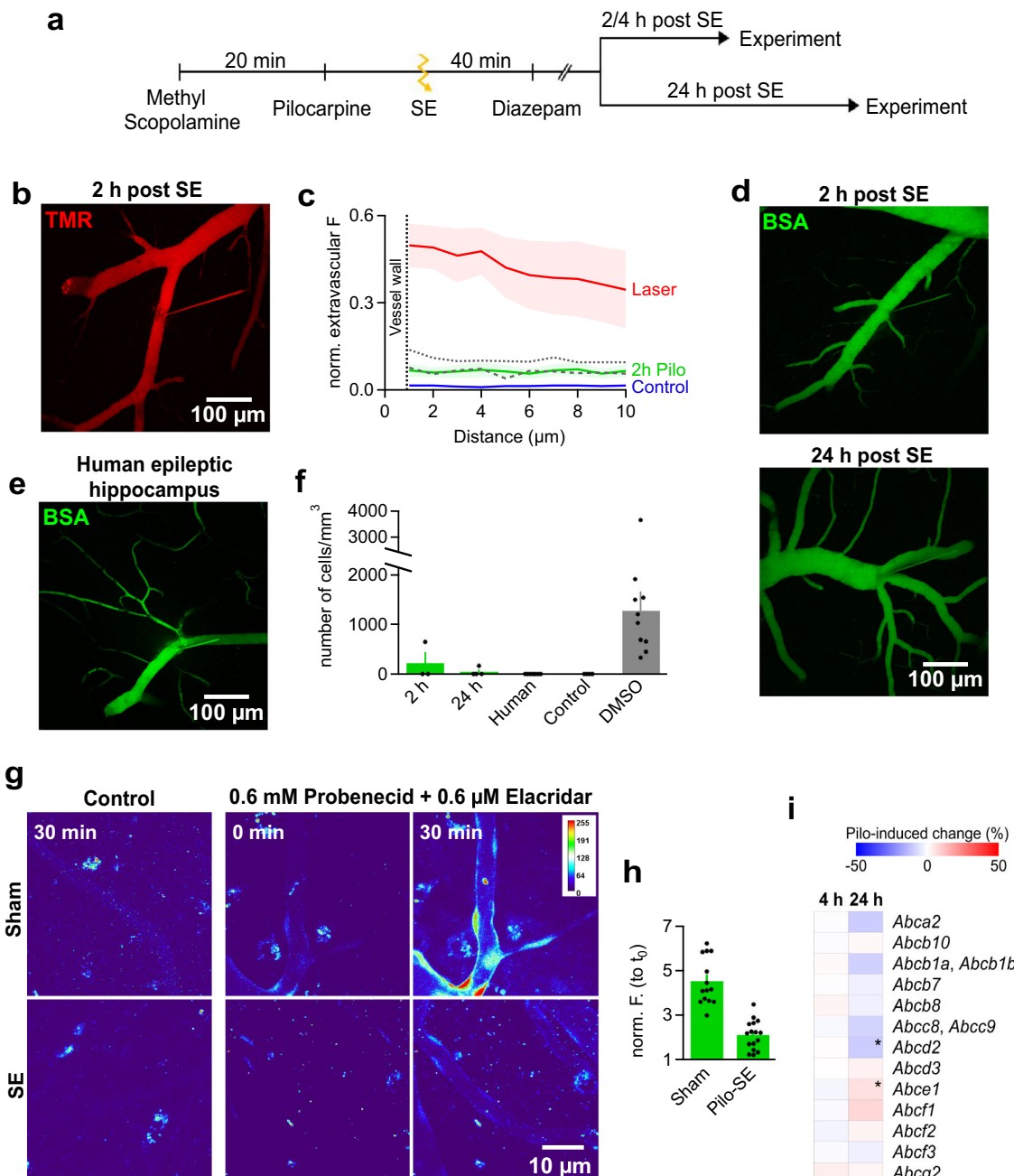

**Fig. 7 | BBB in temporal lobe epilepsy. a** Timeline of pilocarpine experiments. **b** MIP after TMR injection at 2 h post SE, quantified in (c). **c**, Fluorescent profiles across walls of capillaries. Comparison across conditions, mannitol (dotted line), DMSO (dashed line). **d** MIPs of BSA-Alexa488 (BSA)-injected brain microvasculature assessed 2 h (top) and 24 h (bottom) post SE. **e** MIP of BSA-filled microvasculature within a human hippocampal brain slice. Image is representative of 7 injections performed in hippocampi of 5 TLE patients who underwent neurosurgical treatment of their epilepsy. **f** Quantification of BSA-Alexa488-loaded macrophages in the murine epileptic condition at 2 h (*n* = 3) and 24 h (*n* = 4). Data are compared to those obtained from human (*n* = 7), control (*n* = 4) and DMSO (*n* = 10) samples. **g**, MIPs of capillaries of sham (top row) and 2 h post SE murine slices (bottom row) without preincubation with blockers (control) at $t_{30}$ and under full ABCC1/partial

ABCB1 inhibition with 0.6 mM probenecid and 0.6 µM Elacridar, respectively, at $t_0$ and $t_{30}$, quantified (see h). Images are representative of 4 experiments. **h**, Calcein signal at $t_{30}$, normalized to that at $t_0$ (norm. F.), 11 injections/ 6 animals, $n_{ROI}$ of control = 14, $n_{ROI}$ of SE = 16. **i**, Heat map of percentage change in protein levels for ABC transporter proteins at 4 and 24 h after pilocarpine treatment of mice relative to sham treatment, as determined by mass spectrometry. Percentage is an average of 4 measurements (*n* = 4 mice per condition). "*" designates significant changes (*P* value < 0.05). The *P* values denoting the significance of Abcd2 and Abce1 transporters relative to sham treatment were 0.003574 and 0.047767, respectively. Source data to panels of Fig. 7 are provided as a Source Data file. Data are presented as mean values +/- SEM.

replicate the disease in vitro, which is not possible if the disease itself is not fully understood or complex.

In vivo models on the other hand work with the native BBB and can be combined with animal models and represent an optimal combination in this respect. However, in vivo models systematically

differ from our model in applicability and possible read-outs. In vivo (injection) models broadly fall into two classes: a) sacrificing the experimental animal before analyzing uptake of the test substance into the brain and b) analyzing *uptake* in vivo in the living organism (e.g. 2 P in vivo imaging, PET imaging[82]). For the former, usually a

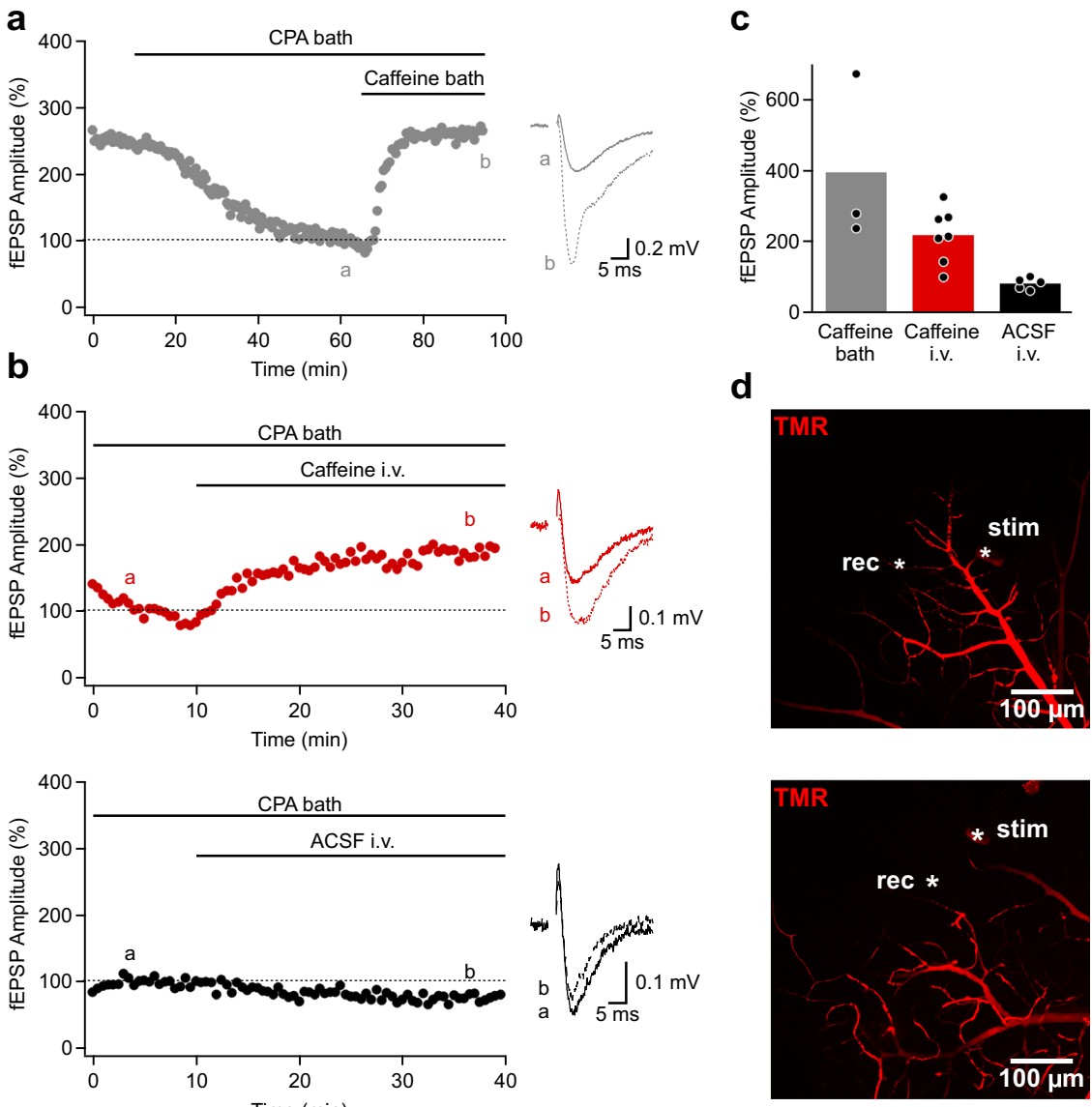

**Fig. 8 | Monitoring the pharmacological effect on neuronal responses after drug passage through the BBB. a** Time course of an example experiment of fEPSP recordings in the hippocampal area CA1. The adenosine A1-receptor CPA reduces synaptic responses. This effect is antagonized by co-application of caffeine. Inset shows recordings (average of 5 traces) from times indicated by lower case letters. **b** Top panel: Intravascular (i.v.) application of caffeine also antagonizes the effect of CPA (pre-applied) and increases synaptic responses. Note the different time scale of the x-axis when compared to **a**). Bottom panel: Control intravascular injection of ACSF does not alter synaptic responses. **c** Summary of percentage fEPSP amplitude changes after caffeine bath application ($n = 3$ from 2 injections/2 mice), intravascular caffeine application ($n = 5$ from 3 injections/4 mice) and intravascular ACSF application ($n = 5$ from 3 injections/3 mice). Bar graph shows mean ± SEM. **d** MIPs of caffeine (top) and sACSF (bottom) injected vascular tree. The asterisks denote the positions of the stimulating (stim) and recording (rec) pipettes. Images are representative of 10 experiments. Source data to panels of Fig. 8 are provided as a Source Data file.

biochemical or histochemical readout of BBB function is used and permeating test molecules are quantified per volume or weight of brain tissue, but the extracellular concentration, relevant for neurons and glial cells, remains imprecisely known as it depends on the subcellular distribution of the test molecule (e.g. intra- vs extracellular, membrane- vs. lipid-bound). Furthermore, it is difficult to exclude or identify changes of test molecule distribution caused during fixation of the tissue or occurring post-mortem. Of all approaches, the second class of in vivo models employing high resolution 2 P in vivo imaging as read-out have the closest and most realistic look at the BBB. However, due to movements of the living brain and strong light scattering at high imaging depths, it is difficult to obtain very high image resolution (e.g. for quantifying fluorescence of EC membranes) and certain experiments or experimental combinations are not possible or very demanding. For example, some brain regions (e.g. medulla oblongata, thalamus), some organs (e.g. heart) and early developmental stages (e.g. embryonic) are not readily accessible for in vivo imaging. In addition, while local imaging is the rule, intravenously applied test molecules will distribute in the whole circulation and reach all parts of the brain. This makes it difficult to monitor where and when substances enter the brain (veins, capillaries, arteries, choroidal membranes, lymphatic system) and to control the effective concentration in cerebral blood vessels near the imaging site as excretion, metabolism and binding to blood components are hard to control confounding factors. Finally, combining imaging in vivo with compound and drug injection as well as cellular electrophysiology for physiological recordings is very challenging.

ISMICAP offers precise control of both luminal (capillary perfusion) and parenchymal (bath perfusion) concentrations of drugs and tracers and constant concentrations are maintained as long as the vascular system is perfused. This made it possible to quantitatively compare the accumulation of calcein in the presence of several ABC-transporter blockers and to show that ABCC1 shows a much weaker calcein-AM extrusion rate compared to ABCB1. The precise control of the concentrations on both sides of the BBB together with subcellular optical or cellular physiological recordings offer the unique opportunity to obtain a dose-effect titration directly at the blood barrier as we illustrated for the case of caffeine.

In a slice preparation, the brain tissue including the vasculature and the NVU are fed with oxygen and nutrients (such as glucose) via the bath perfusion medium. This provides the experimental freedom to modify the luminal solution, such as removing blood cells or changing the chemical or cellular composition, without the danger of damaging the NVU or the nervous tissue as in vivo imaging approaches would be faced with. This allowed us to flush out cellular components from the vasculature and apply the fluorescent membrane dye FM1-43. Only because cellular components were removed, we obtained a clear and unobstructed view of the stained EC boundaries and could visualize the shape and boundaries of living ECs and quantify their surface area. This is an important parameter to assess the cellular transport load at the BBB.

Establishing a solution stream through the vessel system of brain slices also advances the applicability of slice experimentation, for example to physiological studies. Responses of the NVU unit depend on the actual diameter and/or the intracapillary pressure, but conventional brain slice preparations lack capillary perfusion resulting in a collapse of capillaries and venules. Lack of capillary perfusion in brain slices may be even more important for studying potassium buffering and clearing by astrocytes in brain slices. While brain slices are optimally suited to accurately measure the local potassium concentration, to stimulate, manipulate and record neuronal activity and the uptake capabilities of astrocytes, one important component of potassium homeostasis is missing: the removal of potassium through astrocytic endfeet into the circulation. ISMICAP now experimentally re-generates this potassium sink by clamping intracapillary potassium levels and allowing astrocytes to release the neuronal potassium they have been collecting in the parenchyma during periods of neuronal activity into the vascular lumen.

ISMICAP also comes with certain limitations. The controllability of luminal and parenchymal solutions may also be a disadvantage when studying effects of endogenous molecules on the BBB. For ISMICAP, the endogenous solutions are removed, whereas they are maintained in in vivo models. Further, while the slices used for the approach presented here had been in contact with the circulation and the rest of the organism, they are isolated during the experiment. Thus, any acute effects of e.g. the composition of the blood on the BBB are (almost) impossible to study in a slice model. Finally, the advantage of working with native tissue can represent a disadvantage if the goal is to study the effect of differences in the composition of the NVU or surrounding tissue. While in vitro models do allow an engineering of the experimental tissue and provide a certain level of control of the cellular environment this is not possible with acute brain slices and also difficult with cultured slices.

The usage of 2 P microscopy is key to the experimental flexibility of ISMICAP. 2 P microscopy vs. other excitation methods is necessary to follow the arbors of the vascular tree, which penetrate deeply into the tissue (>100 μm). Only this high-resolution live cell imaging approach enabled us to firstly observe the slow, membrane-delimited diffusion of FM1-43 beyond endothelial tight junctions, the extravasation of wall-resident perivascular macrophages upon strong damage of the BBB and to exclude strong transcytosis of fluorescently-labelled proteins through ECs after DMSO treatment and status epileptics.

The fact that ISMICAP is based on a slice preparation delivers the advantage of universality. Almost any tissue and organ from many species can be cut into slices and kept in a recording chamber for micro-perfusion of the contained capillaries. In this study, we demonstrate this versatility by examining BBB properties of an animal model of epilepsy and by analyzing the native human BBB in a disease context supporting the translational value of ISMICAP.

## Methods

All experiments with specimens of murine origin were performed in accordance with the guidelines of the University of Bonn Medical Centre Animal-Care-Committee. All efforts were made to minimize pain and suffering and to reduce the number of animals used, according to the ARRIVE guidelines. Animals were provided with nesting material, water and food ad libitum and kept under control of an alternating 12-hour light-dark-cycle (light-cycle 7am–7pm), in a temperature ($22 \pm 2\,°C$) and humidity ($55 \pm 10\%$) controlled environment.

Hippocampal tissue of pharmacoresistant TLE patients was neurosurgically removed for seizure control in the context of the local epilepsy surgery program[71]. Tissue from 5 individuals were analyzed (3 males 27, 61, 21 years old, 2 females 40 and 14 years old). For human tissue sampling, it was ensured that each patient gave written consent for the scientific use of her or his tissue and permission was obtained from the Ethics Committee of the University of Bonn. All procedures adhered to national and institutional ethical requirements and were conducted in accordance with the Declaration of Helsinki.

### Manufacturing of injection-pipettes
Glass capillaries were horizontally pulled (Sutter Instruments) and subsequently underwent abrasive treatment at an angle of approximately 30–40° (World Precision Instruments). Pipettes with tip-openings >3 μm or contaminated with debris were excluded from experiments. Randomly selected pipettes were visualized via scanning electron microscopy (Hitachi). Samples were carefully mounted on modelling clay, sputter coated with gold (Quorum Technologies) and imaged at 5–10 kV under high vacuum conditions. Secondary electrons were detected with a SE-detector.

### Acute brain slices preparation
Post-natal 25–35-day old C57bl/6NCrl male mice (Charles River Laboratories) or mice from the pilocarpine-induced mesial TLE model were used. After anesthetizing with isoflurane, mice were decapitated, the brain was removed from the skull and rapidly submerged into ice-cold modified artificial cerebrospinal fluid (mACSF) containing (in mM): 87 NaCl, 2.5 KCl, 1.25 $NaH_2PO_4$, 7 $MgCl_2$, 0.5 $CaCl_2$, 25 $NaHCO_3$, 25 glucose, and 75 sucrose (saturated with 95% $O_2$/5% $CO_2$). Horizontal cortical or hippocampal slices (300 μm) were prepared on a vibratome (HM650 V, Thermo Fisher Scientific, Waltham, USA). Subsequently, slices were quickly transferred to a submerged chamber containing mACSF at 35 °C for 30 min before stored at room temperature in standard artificial cerebrospinal fluid (sACSF) containing (in mM): 124 NaCl, 3 KCl, 1.25 $NaH_2PO_4$, 2 $MgCl_2$, 2 $CaCl_2$, 26 $NaHCO_3$, 10 glucose (saturated with 95% $O_2$/5% $CO_2$). Experiments were not started earlier than 30 min after dissection. During the experiment, perfusion solutions were constantly aerated with a mixture of $O_2$/$CO_2$ (95%/5%) and the pH was verified before each experiment to ensure sufficient pre-bubbling to reach the equilibrium pH of 7.4 (at RT) (Suppl. Figure 1). During perfusion, the beakers were partially covered to improve gassing of the solution. In general, slices were used for the span of 6–8 h following their preparation and the BBB properties explored in this study appear stable within that period.

Intravascular injections were performed on brain slices in a submerged specimen chamber of a two-photon laser scanning microscope (Suppl. Figure 1a, b) either from Nikon (A1R MP), Scientifica (2PIMS-PMT-20) or Bruker Scientific (Olympus BX51WI), each equipped with one or two chameleon vision II lasers (Coherent) and with NIS-Elements AR 4.13.01, Scanimage r3.8 or Prairie View 5.4 software for scanning and acquisition, respectively. Slices were constantly perfused with carbogenated sACSF at a flow rate of 1–2 ml/min. Perfusion-pipetted were filled with 5 μl of the respective test solution, inserted into a pipette holder mounted to a micromanipulator (Luigs & Neumann) and connected to a pressure container regulated by a patch perfusion system (ALA Scientific Instruments) (Suppl. Figure 1c, d). Under optical control, pipette tips were lowered into the tissue and approached to a venous vessel membrane in 20–50 μm depth. Next, vessel membranes were punctured by the cannula-like pipette tip. Subsequent application of 100 mmHg pressure flushed the vascular system with the pipette solution. Pressure on the injection-pipette was kept constant throughout the experiment. Successful injections were indicated by absence of intense dye leakage around the injection point (IP).

Biocytin tetramethylrhodamine (TMR, Sigma Aldrich) was perfused into cortical and hippocampal vessels at a concentration of 100 μM in sACSF. Single frame time series over 20 min (1 frame/5 s), as well as z-stacks (plane distance 0.25–1 μm) were recorded. If non-fluorescent molecules were injected into the vasculature, TMR was added to the injection solution at a concentration of 100 μM as an indicator for a successful piercing and perfusion.

Time to reach steady state was assessed on time series. For quantification, fluorescence intensities were determined within regions of interests (ROIs) positioned at nine distances to the IP. 'Steady state' of intravascular dye fluorescence was defined 95% of the plateau level of fluorescence reached at >5 min (cf. Figure 1c). The elapsed time from starting pressure application to reaching this at each of the nine distances is plotted in Fig. 1d and was called "time to steady state".

Spatial intravascular gradients of TMR were assessed by intensity profiles taken from maximum intensity projections (MIPs). Two profiles were taken along the punctured primary vessel starting at the IP until a distance of at least 150 μm (Fig. 1f). Additional intensity profiles were taken from branches of that primary vessel. Profiles from these branches were groups into three categories according to their distance from the IP (<100, <200, >200 μmcf. Figure 1e). The profiles of each category are shown in 3 separate panels in Fig. 1 f. The color of the lines in all panels codes the order and the diameter (above or below 10 μm) of the measured branch as illustrated by the cartoon in Fig. 1e. All profiles were normalized to the fluorescence reached at the IP.

Estimation of the surface area of individual ECs. Based on the fact that only one longitudinal border of the EC was visible, it could be assumed that an EC in a capillary occupies a quarter of the vessel's 3D surface area. Accordingly, the calculations were adjusted by multiplying the measured 2D surface area of the EC by the factor $[(\pi \times d \times l)/4 \div (d \times l/2)] = \pi/2$, where d and l are the diameter and length of the EC-contained section of a vessel. ECs in venules were fully visible and the measured area was not scaled. Analysis of fluorescent images was performed with the ImageJ platform (1.51 to 1.53).

Dyes were obtained from: 7-hydroxycoumarin-3-carboxylic acid (7HCC), Sigma Aldrich; sulforhodamine 101 (SR101), R&D Systems; Lycopersicon Esculentum Lectin-DyLight 488 (Tomato lectin), Thermo Fisher Scientific; FM1-43/ FM1-43fx, Thermo Fisher Scientific; Rhodamine123 (Sigma Aldrich); MitoTracker CMTM (MitoTracker), Thermo Fisher Scientific; calcein-AM, Thermo Fisher Scientific; Neurotrace 500/525 (NT500/525), Thermo Fisher Scientific; Calcein red-orange (red-shifted calcein), Thermo Fisher Scientific; Hoechst33342, mannitol, Sigma-Aldrich; bovine serum albumin-Alexa Fluor 488 conjugate (BSA-Alexa488), Invitrogen, DMSO (Carl Roth).

MIPs of venules and capillaries 30 min after perfusing the vasculature with FM1-43 or tomato lectin were used for the surface area measurement of ECs. In venules, ECs can be manually fully outlined, while they could only be partially outlined in capillaries as one of their longitudinal borders was hidden in the side wall of the capillary. Therefore, it could be assumed that an EC in a capillary occupies a quarter of the vessel's 3D surface area (if it would be more than a quarter, a second border would have appeared) and part of the EC remains undetectable in the projection. To correct for this error associated with the projection, the calculations were adjusted by multiplying the measured 2D surface area of the EC by the factor $[(\pi \times d \times l)/4 \div (d \times l/2)] = \pi/2$, where d and l are the diameter and length of the EC-contained section of a vessel, respectively.

Adult male C57Bl/6-N mice received a dose of scopolamine methyl nitrate (1 mg/kg, s.c.; Sigma Aldrich) 20 min prior to the administration of pilocarpine hydrochloride (335 mg/kg, s.c.; Sigma Aldrich). Forty min after SE onset, mice received 4 mg/kg s.c. diazepam (Ratiopharm). Control animals were treated identically, but received saline instead of pilocarpine. After pilocarpine injection, animals exhibit several stage 3 (severe seizures with rearing without falling) and stage 4 seizures (severe seizures with rearing and falling)[83]. Behavioral SE was clearly identified using a modified seizure scheme, with persistent immobilization, postural loss and sustained generalized convulsions[84]. Among pilocarpine-injected animals, only those that developed SE (SE-experienced) were further used for analysis. For calcein accumulation assay, it was conducted as previously described. However, slices were pre-incubated and constantly perfused with 0.6 mM probenecid and 0.6 μM Elacridar.

## Preparation of hippocampal tissue from pilocarpine-treated mice for proteomics analysis

Male C57Bl6/N mice were either sham- or pilocarpine injected as described above. Both hemispheres of the hippocampus were extracted at 4 or 24 h post-pilocarpine injection and frozen on dry ice. Sixteen mice were injected allowing for four mice per time point. The hippocampal tissue was homogenized in 500 μl PBS and 125 μl lysis buffer was added consisting of 2% sodium dodecyl sulphate, 50 mM trisaminomethane/HCl pH 7.4, 2 mM ethylene glycol-bis(β-aminoethyl ether)-N, N, N0, N0-tetraacetic acid, 2 mM ethylenediaminetetraacetic acid, Complete Protease Inhibitor Cocktail (Roche), 2 mM phenylmethylsulfonyl (Sigma), 5 mM NaF (Sigma), 2 mM beta-glycerophosphate (Sigma), Phosphatase Inhibitor Cocktail 2 (1:1000) and PhosSTOP (Roche). Lysates were incubated at 85 °C for 10 min, sonicated for 3 × 10 s, frozen and lyophilized. Samples were resuspended in 100 μL of 10 mM tris(2-carboxyethyl) phosphine, reduced for 10 minutes at 85 °C and alkylated in 25 mM iodoacetamide at 22 °C. Proteins were precipitated from the samples using chloroform-methanol extraction and air dried. The precipitates were dissolved in 20 μL solution containing 7.8 M urea, 50 mM triethylammonium bicarbonate and 5 μg of Lys-C (FUJIFILM Wako Pure Chemical Corporation) for an 8 h digestion at 25 °C, then diluted with a solution containing 170 μL 100 mM 4-[2-hydroxyethyl]-1-piperazineethanesulfonic acid (HEPES) (pH 8) and 5 μg TrypZean trypsin (Sigma Life Sciences) for subsequent digestion at 37 °C for 4 h. The trypsin digestion was twice repeated[85]. Two TMT-10plex kits (Thermo Fisher Scientific) were used to label 400 μg of peptide to generate two 8-plex sets, one for each time point. The 4 h TMT sample set consisted of: sham replicate 1, 128 N; sham replicate 2, 128 C; sham replicate 3, 129 N; sham replicate 4, 129 C; pilocarpine replicate 1, 130 N; pilocarpine replicate 2, 130 C; pilocarpine replicate 3, 131 N; pilocarpine bio-replicate 4, 131 C. The 24 h TMT sample set consisted of: pilocarpine replicate 1, 128 N; pilocarpine replicate 2, 128 C; pilocarpine replicate 3, 129 N; pilocarpine replicate 4, 129 C;

sham replicate 1, 130 N; sham replicate 2, 130 C; sham replicate 3, 131 N; sham replicate 4, 131 C (lot numbers TB266076 and TA265136). The TMT labelled samples were diluted to 2 mL with 0.1% trifluoracetic acid and desalted using a Sep-Pak tC18 3cc Vac cartridge (200 mg sorbent, Waters), then fractionated by hydrophilic interaction liquid chromatography using a Dionex UltiMate 3000 HPLC system (Thermo Fisher Scientific) controlled by Chromeleon 6.8 software with a TSKgel Amide-80 1 mm inside diameter x 250 mm long column (Tosoh Bioscience). Samples were injected in 150 μL in 90% acetonitrile, 9.9% water, 0.1% trifluoracetic acid (HILIC Buffer A) at a flow rate of 60 μL/min with buffer A at 100% for 10 min. The gradient was from 100% HILIC buffer A 100% to 40 % HILIC buffer B (99.9% water and 0.1% trifluoracetic acid) for 35 minutes at a flow rate of 50 μL/min, then to 80% buffer B for 3 minutes then back to 100% buffer A for 12 minutes. Sample fractions were collected in 60 s intervals into a 96-well plate using a Probot (LC Packings) controlled by μCarrier 2.0[85].

LC-MS/MS was performed on each fraction using an Ultimate 3000 RSLC nano system and Q Exactive Plus hybrid quadrupole-orbitrap mass spectrometer (Thermo Fisher Scientific). The TMT sets were pre-screened for ABC transporter peptides by data-dependent acquisition as described previously[85]. Then, the identified ABC transporter peptides were targeted for re-analysis using an inclusion list, incorporating precursor mass-to-charge ratio, precursor charge state and elution time. A total of 124 peptide sequences corresponding to twenty-five ABC transporter proteins and two standard cytoskeletal proteins (Dynein heavy chain 1 and alpha-tubulin), as loading controls, were targeted by LC-MS/MS. Samples were loaded directly onto a 300 × 0.075 mm column packed with ReproSil Pur C18 AQ 1.9 μm resin (Dr Maisch, Germany). A column oven (PRSO-V1, Sonation lab solutions, Germany) integrated with the nano flex ion source (Thermo Fisher Scientific) maintained the column at 50 °C with an electrospray operating at 2.3 kV. The S lens radio frequency level was 60 and capillary temperature was 250 °C. A The samples were loaded in 5 μL at 300 nL/min for 25 min, then the gradient was from 5% to 25% buffer B in 74 min, then to 35% buffer B in 8 min and to 99% buffer B in 1 min, held at 99% B for 2 min, then to 1% B in 1 min and held at 1% B for 8 min. MS/MS scans were acquired using parallel reaction monitoring at a resolution of 35,000 full width at half maximum with an isolation width of 0.8 m/z for a maximum ion time of 400 ms and automatic gain control target of 50,000 counts.

## Processing of mass spectrometry data

LC-MS/MS data output was processed with MaxQuant v1.6.7.0[86]. The Mus musculus reference proteome was downloaded March 14, 2022 and contained 63,628 canonical protein isoform entries. Reporter mass tolerance was 0.005. Filtering by product ion fraction was 0.6. Variable modifications allowed were oxidation of Met, acetylation of the protein N-terminus and deamidation of Gln/Asn. Carbamidomethylation of Cys was a fixed modification. The enzyme specificity was Trypsin/P and three missed cleavages were allowed. The minimum peptide length was 6 and the maximum peptide mass was 6,000 Da. Modifications were not allowed in the protein quantification. Unmodified counterpart peptides were discarded. Second peptide were allowed. All other MaxQuant parameters were default. The proteinGroup text file from the MaxQuant output was processed in Microsoft Excel. The intensities of the ABC transporter proteins, dynein heavy chain 1 and alpha-tubulin were log2 transformed across the 8 channels for both the 4 h and 24 h sets. The ABC transporter intensities were then normalized using the combined dynein heavy chain 1 and alpha-tubulin intensities. The normalized values were then statistically compared using GraphPad Prism 9. Two comparisons were made: i) 4 h pilocarpine treated against 4 h sham treated and ii) 24 h pilocarpine treated against 24 h sham treated. Proteins with <2 unique peptides at both 4 h and 24 h were excluded from the analysis. A t-test

was performed, which was adjusted for multiple comparisons using a false discovery rate of 1%. The heatmap of the percentage change was generated using Morpheus v1 (https://software.broadinstitute.org/morpheus). The proteinGroups file and annotated MS/MS spectra for the ABC transporter peptides are supplied as supplementary material (MS files.zip). The mass spectrometry proteomics data have been deposited to the ProteomeXchange Consortium via the PRIDE [1] partner repository with the dataset identifier PXD037454 and 10.6019/PXD037454.

BSA-Alexa488 and FM1-43 injected brain slices were post-processed after time-lapse recordings by overnight incubation at 4 °C in a 24-well plate with 4 % paraformaldehyde solution (PFA; Sigma Aldrich). Control brain slices (500 μm) were incubated overnight at 4 °C in 4% PFA immediately after slicing. Afterwards, all slices were washed thrice for 10 min with TRIS-buffered saline (TBS; 0.1 M). The slices were embedded in 3% agar (Sigma Aldrich) and trimmed to 50 μm thin sections. For cell specific staining, slices were incubated overnight with primary antibodies in TBS at 4 °C (in 0.06–0.1% Triton-X). After 3 10-min washing steps with TBS, slices were incubated with secondary antibodies (in 0.06–0.1% Triton-X) for 3 h at 35 °C then washed as described. For nuclear labelling, the slices were incubated with Hoechst33342 (20 μg/ml; Sigma Aldrich) for 20 min then washed. The slices were mounted on slides and subsequent image acquisition was performed by confocal laser scanning microscopy (Nikon).

The following primary antibody were used: TMEM119 (rabbit, 1:100; Abcam), VE-cadherin (rabbit, 1:200; Abcam), F4/80 (rat, 1:100; Novus Biologicals). Secondary antibodies: Alexa 488-conjugate (anti-rabbit, 1:100; Abcam), rhodamine red X-conjugate (antirabbit, 1:200; Jackson Immune Research Laboratories), Alexa 568-conjugate (antirat, 1:200; Invitrogen).

For the combination of micro-perfusion and field potential recording, first, stimulation and recording electrodes (glass pipettes) were lowered into the stratum radiatum of the hippocampal CA1 region. Next, the perfusion pipette was lowered into the tissue and venules were perforated near the hippocampal fissure. Field potentials signals were amplified with a field potential amplifier EXT-02F/2 (npi electronic GmbH, Tamm, Germany) and a postamplifier BF-48DGX (npi electronic GmbH, Tamm, Germany). Signals were filtered at 3 kHz (by postamplifier), digitized with a sampling frequency of 10 kHz (NI USB-6229 National Instruments, Austin, TX, USA) and recorded using the IGOR Pro software (version 5–7, Wavemetrics, Portland, OR, USA). fEPSP amplitudes were determined from the baseline prior to stimulation to the maximum of the negative deflection. Amplitudes were normalized (100%) on the level achieved in CPA (cf. Figure 8).

Data are represented as mean ± standard error of the means (SEM). For comparison of two groups, two-tailed Student's *t*-test was used. For comparison of more than two groups, one-way analysis of variance (ANOVA) test was applied (two-tailed) coupled with Tukey's post *hoc* test for correction of pairwise multiple comparisons. Differences were accepted as statistically significant at $P < 0.05$. Statistical analysis was conducted using GraphPad Prism 9.0.2 (GraphPad Software, CA, USA). All statistical tests applied were two-sided.

## Reporting summary

Further information on research design is available in the Nature Portfolio Reporting Summary linked to this article.

## Data availability

Source data are provided with the paper. Mass spectrometry data generated in this study has been deposited and made publicly available at the PRIDE database with the accession code PXD037454. Source data are provided with this paper.

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

## Acknowledgements

Supported by the Deutsche Forschungsgemeinschaft (SFB 1089 to AB, DD, SS, SPP1757 to DD and SS, INST1172 15, DI853/3-5&7 to DD, SCHO 820/4-1, SCHO 820/6-1, SCHO 820/7-1, SCHO 820/5-2 to SS, FOR 2715 to AJB), the BONFOR program of the Medical Faculty of the University of Bonn (to ASH, JP, AJB, SS, DD), the European Union's Seventh Framework Program (FP7/2007-2013) under grant agreement n°602102 (EPITARGET; AJB, SS), Bundesministerium für Bildung und Forschung (the EraNet DeCipher to AJB), Verein zur Förderung der Epilepsieforschung (to SS & AJB), CONNECT-GENERATE (FKZ01GM1908C to AJB), Else Kröner-Fresenius-Foundation (Promotionskolleg 'NeuroImmunology' to AJB, 2016_A05 to JP), The Alexander-von-Humboldt Foundation via a Georg Forster postdoctoral fellowship (to ASH).

## Author contributions

A.S.H. and P.S. carried out experiments in brain slices and analyzed and graphed data. J.P. and A.J.B. provided status epilepticus model mice and input thereof. M.E.G. and M.HS. performed proteomics, mass spectrometry and related bioanalyses. N.V. assisted with Dodt scan acquisition. D.D. and A.L. conceived the study and designed the experiments. A.S.H., P.S., S.S., M.E.G., A.L., A.J.B., J.P., N.V. and D.D. wrote the manuscript.

## Funding

## Competing interests

The authors declare competing interests.
