## [Peer Review File · Nature Communications]

Subcellular analysis of blood-brain barrier function by
micro-impalement of vessels in acute brain slicesREVIEWER COMMENTS

Reviewer #1 (Remarks to the Author):

The manuscript entitled "High-resolution analysis of blood-brain barrier function by micro-impalement of capillaries in brain slices" by Steinlein and colleagues describes a new method to study the integrity of the blood-brain barrier in acute brain slices from mice and human samples. The technique developed by the Dietrich lab shows interesting avenues but requires additional validation. The application of the technique is very broad in the manuscript. Instead, it would make more sense to investigate a clear biological question. In addition, some of the authors' conclusions are too strong and not fully supported by their data.

Major points:

1. The authors claim that the integrity of the blood-brain barrier is maintained in acute brain slices after intravenous injection of TMR (Fig. 1). It is important to further characterise this point in their model system. How long is the tightness of the barrier maintained? Is there a point at which the vessels start to leak? For which tracer sizes is the method suitable? The authors should assess the tightness of the BBB in the acute slices for a broader representation of tracer sizes.
2. The results with FM1-43 are very interesting and this dye seems to be a potential tool for labelling the brain endothelial membrane *in vivo*. However, the authors' statements are too optimistic about its use without deeper validation. FM1-43 is often used to study exocytosis/endocytosis, how long does the dye remain on the membrane of brain endothelial cells? Could the authors demonstrate that there is no endocytic transport of the dye? Are transcytotic vesicles detectable? In addition, the authors claim that the dye staining is useful to determine the size of the cells. They measured the area of the ECs, but the cell volume would be more representative of the morphology that the cells adopt in the different vessels. The measurements of cell sizes under the sample conditions should be supported by comparison with other methods, i.e. immunofluorescence staining of membrane proteins such as VE-cadherin or by 3D reconstruction of electron microscopy.
3. The authors suggest that FM1-43 may be useful for delivering molecules in the brain by passive diffusion in the cell membrane. Is FM1-43 detected in neurons or astrocytes after a certain time following intra-luminal injection?
4. Rhodamine123 (used in Fig. 3) is a small dye (about 380KDa) that is extruded from the BBB by efflux transporters. This effect is shown in the model after inhibition of the ABCB1 transporter. However, it is surprising that the authors do not find any extravasation of the dye in the brain after inhibition of efflux transport. Please explain.
5. The images in Fig. 3 with the injection of Calcein after ABC transporter inhibition seem to indicate that pericytes could also take up the dye. The authors should evaluate how much of the fluorescence signal corresponds to pericytes or endothelial cells.
6. The authors claim that the effects of DMSO on BBB disruption are mediated by the weakening of the tight junctions. This conclusion is not accurate and not based on data. Can the authors show that the tight junctions are in fact defective? Or that transcytosis is not increased? Is DMSO causing certain toxicity to the endothelial cells?
7. The statements about phagocytosis mediated by perivascular macrophages are too strong and not

fully substantiated. First, the distinction between microglia and macrophages is more complex and requires more careful classification, i.e. F4/80 is not an exclusive marker for macrophages. Second, the distribution of perivascular macrophages could be different depending on the type of vessels, did the authors observe differences depending on vessel type? Third, the authors say that perivascular macrophages are recruited after DMSO injection, but the "recruitment" mechanism and their "migration" are not explained.

8. The introduction of the epileptic focus is abrupt and not well integrated into the main text. In addition, the results do not appear to be fully consistent, as the authors state that dye extravasation was observed in only one of the animals. The results in human samples also showed no extravasation, but they were performed in patients who had recovered from seizures. Despite the negative results, the authors conclude that they could be caused by a slight weakening of the tight junctions. The authors should be cautious about speculating unless they can show solid evidence to support their hypothesis.

Minor points:

1. Some of the Figure fields are not described in the main text.
2. In Fig.3, it would make more sense to show the cartoon of the results according to the experimental data.
3. The use of diazepam in Fig. 5 is not explained in the main text.

Reviewer #2 (Remarks to the Author):

In this methods paper, Steinlein et al. describe establishing an in situ microperfusion method in brain slices that allows assessing blood-brain barrier function, specifically transporter function and barrier leakage. The idea of such a microperfusion is not new and has been discussed in the brain barriers field for decades. To the authors' credit, they now established this method. The authors describe perfusing microvessels in mouse brain slices using a set of fluorescent dyes to visualize the vessel lumen and membranes with 2P microscopy. They also used fluorescent efflux transporter substrates and transporter inhibitors to visualize transport activity. In another set of experiments, the authors demonstrate that their method can be used to visualize barrier leakage. Then, the authors used brain slices from mice after status epilepticus and from patients who underwent epilepsy resection surgery and observed mild leakage. Lastly, the authors combined microperfusion with neuroelectrophysiological measurements showing this approach can be used to measure pharmacological effects.

Comments:

This MS is reasonably well-written, but English grammar, language and style need to be revised. For example, the authors use many terms that are simply not used in the English language. Suggest having a native English speaker carefully revise the MS.

Given that the authors describe that they could only puncture larger vessels, but not capillaries (lines 162-165), the MS title seems inappropriate.

This methods paper provides an interesting new technique to study the blood-brain barrier, but its significance is overstated. Due to the relatively complex setup, it is unlikely this method will be used much. In addition, many researchers in the field are currently using in vivo 2P imaging that accomplishes the same (minus electrophysiology) in a more realistic in vivo setup but without the technological challenges shown here. This should be addressed in the Discussion.

The authors criticize existing methods in the field, including in vitro cell culture, isolated capillaries, and in vivo 2P imaging approaches to assess barrier function, but then only briefly brush over the shortcomings of their model. For example, while the authors correctly point out how in vitro BBB models do not reflect the in vivo situation, they seem to forget that their own model is not an in vivo model but is based on brain slices using artificial buffers, which does not reflect the in vivo situation. In this regard, the authors overinterpret some of their findings and inappropriately overstate the relevance of their approach while at the same time making invalid comparisons with existing models and approaches – these are apple-orange comparisons that are no valid because these models are used for different purposes. The authors should also not forget that what model is used is based on what question is asked. Lastly, other models have been quite useful over many decades but, just like the approach presented here, they are models and at the end of the day: all models are wrong, but some are useful (George Box) – and this certainly includes the model presented here.

The M&M section lacks significant experimental detail to properly assess some of what is reported. For example, to improve this MS, the authors should provide pictures of the setup, provide detailed imaging settings, provide details of the SE inductions in mice. Were mice really dosed s.c. or i.p. with methyl-scopolamine, pilocarpine, diazepam? (i.p. is what is mostly used in the literature), provide a video of the mice seizing, provide antibody concentrations (not dilutions).

Major concerns:

- Figure 5 shows scopolamine – are the authors sure they used methyl-scopolamine and not scopolamine? Even though chemically, the difference is only a methyl-group, this would have significant consequences for the entire study. Specifically, scopolamine is a cholinergic antagonist that, amongst others, is used to generate a chemically-induced dementia model and this effect is based on scopolamine's ability to cross the BBB. The methyl-group in methyl-scopolamine, on the other hand, results in a quaternary ammonium group, resulting in a positively charged molecule that does not cross the BBB. And thus, in the pilocarpine SE model, methyl-scopolamine is used to block peripheral cholinergic effects of pilocarpine. The authors need to unequivocally clarify that methyl-scopolamine was used and not scopolamine. The fact that the mistake in Figure 5 was not recognized by any of the authors prior to the submission of this MS raises serious concerns.
- Did the authors keep bubbling the aCSF with O₂/CO₂ during the perfusions? If not, have you measured aCSF pH? If it is not bubbled constantly with CO₂, the pH increases to ~10 within minutes after stopping CO₂ bubbling. This would have a dramatic impact on the experiments/data.
- The inhibitor concentrations used are clinically unrealistic: clinical verapamil blood levels in patients are in the range 0.2-1 microM. Yet, here the authors used 200 microM. What happens with a clinically realistic verapamil concentration? The authors should provide this data. The same is true for probenecid – 1mM is not realistic clinically – and caffeine (300 microM – show data with a realistic caffeine blood

concentration).

- Rhodamine123 is also not a terribly good Pgp substrate, the authors should have used something else.
- Group numbers for the mouse experiments (n=2-3) are too low.
- Provide more details of how fluorescence was measured to obtain Figs. 1, 2.

The Introduction, Results, and Discussion sections need to be revised as they suggest a significant lack of knowledge in the blood-brain barrier field. Examples:

- Lines 51-53: The NVU traditionally includes neurons.
- Lines 69-77: A lot of work has been done over the last 20 years that answers these questions (at least in part). Yet, none of this work has been cited. Recommend you read up on the works of the following groups: Betsholtz, Engelhardt, Miller/Bauer/Hartz, Ronaldson/Davis, Renier, Erturk and others.
- Lines 88-90: The authors seem to be unaware of Kariolis et al. 2020 and Ullman et al. 2020
- Lines 90-93: Existing literature shows that an open barrier does NOT lead to better drug permeation. Epilepsy and brain cancer are good examples: while the barrier is leaky in some areas, overexpression of transporters seems to compensate that (see for the most recent publication deGooijer et al. 2021).
- Lines 94-99: These are all good points but the authors' model has similar weaknesses that are not mentioned.
- Lines 198-199: This is nothing new - it is known that tight junctions limited diffusion within the membrane. In fact, it is the tight junctions that separate the luminal vs. abluminal membranes, thus creating the polarization of the brain endothelium.
- Lines 251-255: This conclusion (Mrp1 is not functional) is not necessarily correct: P-gp levels are much higher than those of Mrp1 and blocking Pgp could simply overwhelm lower Mrp1 levels, and thus, result in staining.

Figure 2D: at the bottom of that microvessel (center of image) there seems to be a cell. Is that a pericyte? If yes, then what is observed is not the abluminal membrane (lines 196-198).

Line 280: using 10% DMSO is a reasonable tool, but clinically unrealistic. Why not use 25% mannitol (as is used in the clinic) to osmotically open the barrier (this is used at OHSU, by the Neuwelt group to open the barrier to deliver chemotherapeutics to brain cancer patients)?

Line 281: BSA-Alexa488 – did the authors purify this product before using it? The reason for purifying BSA-Alexa488 (and similar products) is that there is often unconjugated Alexa488 in these products leading to false data and misinterpretations because observations are based on the dye alone and not the conjugated product due to impurities.

Line 302: DMSO-injured TJs – that's a speculation you have not proof for. DMSO increases membrane fluidity. Is it possible that this may have lead to this observation?

Line 338: SE-triggered changes to the TJs – that's again a speculation that is not backed up by data. While it is true that SE induces loss of TJs, these effects last more than 24h.

Current understanding in the field is that the two most important BBB efflux transporters are Pgp and BCRP – not Mrp1. Why did the authors look at Mrp1 and not BCRP? This seems like a missed opportunity (or another lack of knowledge of the field?) and the authors should provide data showing BCRP-mediated transport.

The authors should add data from experiments assessing Pgp-mediated transport in brain slices from SE mice and/or epilepsy patients. Efflux transporters are upregulated after seizures and this is something the authors should be able to detect with their system.

Reviewer #1 (Remarks to the Author):

"The manuscript entitled "High-resolution analysis of blood-brain barrier function by micro-impalement of capillaries in brain slices" by Steinlein and colleagues describes a new method to study the integrity of the blood-brain barrier in acute brain slices from mice and human samples. The technique developed by the Dietrich lab shows interesting avenues but requires additional validation. The application of the technique is very broad in the manuscript. Instead, it would make more sense to investigate a clear biological question.

We thank the reviewer for his/her appreciation of the potential of our new methodological approach. Following the reviewer's suggestion, we extended the validation of our approach by characterizing BBB permeability with additional tracers (Fig. 2), by testing how long the BBB is stable in slices after slice preparation (Fig. 1g), by showing that mannitol opens the BBB as known from *in vivo* experiments¹ (Fig. 6) and has effects similar to our DMSO treatment. Tomato lectin and VE-cadherin staining were used to verify that FM1-43 indeed outlines endothelial cells (ECs) (Fig. 3c,h), and calcein was shown to accumulate selectively in ECs and not in pericytes (Fig. 4h).

We have also investigated the biological question of BBB alterations in epilepsy in more detail. To this end, we performed a series of additional experiments with epileptic mice and now show for the first time with functional data that P-gp (ABCB1) activity is upregulated 2 h after pilocarpine-induced status epilepticus (SE) (Fig. 7g,h), whereas protein levels are not significantly altered as demonstrated by correlated mass spectrometry (Fig. 7i).

In addition, some of the authors' conclusions are too strong and not fully supported by their data.

We agree and recognize the criticism of both reviewers in this regard and have adapted and toned down our conclusions in the relevant sections (also indicated below).

Major points:

1. The authors claim that the integrity of the blood-brain barrier is maintained in acute brain slices after intravenous injection of TMR (Fig. 1). It is important to further characterise this point in their model system. How long is the tightness of the barrier maintained? Is there a point at which the vessels start to leak? For which tracer sizes is the method suitable? The authors should assess the tightness of the BBB in the acute slices for a broader representation of tracer sizes.

To address the first issue raised by this reviewer, we extended the experimental time window and tested BBB tightness 8 h after slice preparation. We found that the overall tightness of the BBB is maintained but a quantification of dye leakage showed that some TMR leakage can be observed at this later time point (Fig. 1g,h). We conclude

that this acute slice model is useful for a period of up to 8 h – a time frame very similar to that known for the usefulness of slices for electrophysiological recordings. After ~8 h, even neurons in the best slice preparations start to lose optical contrast as a sign of reduced cellular health and become difficult to patch-clamp.

TMR, which was initially used as a tracer, has a molecular weight (MW) of 869 Da. To address the question which size/weight of tracer is suitable for this method we included two additional much smaller tracers in our analysis: Sulforhodamine 101 (SR101, MW 607 Da) and 7-hydroxycoumarin-3-carboxylic acid (7HCC, MW 206 Da) (Fig. 2). Similar to TMR, SR101 does not cross the BBB, but we observed that the small tracer 7HCC permeates the BBB at a rate at least 5-times higher than that of TMR/SR101.

2. The results with FM1-43 are very interesting and this dye seems to be a potential tool for labelling the brain endothelial membrane in vivo. However, the authors' statements are too optimistic about its use without deeper validation. FM1-43 is often used to study exocytosis/endocytosis, how long does the dye remain on the membrane of brain endothelial cells? Could the authors demonstrate that there is no endocytic transport of the dye? Are transcytotic vesicles detectable? In addition, the authors claim that the dye staining is useful to determine the size of the cells. They measured the area of the ECs, but the cell volume would be more representative of the morphology that the cells adopt in the different vessels. The measurements of cell sizes under the sample conditions should be supported by comparison with other methods, i.e. immunofluorescence staining of membrane proteins such as VE-cadherin or by 3D reconstruction of electron microscopy.

To verify that FM1-43 outlines the borders of ECs, we have added two novel experiments. Labelling the surface of endothelial cells (ECs) with tomato lectin produced very similar results to FM1-43 with regard to the intravascular staining pattern and the area measurements based on this marker. In addition, we performed immunostainings against adherens junctions, i.e. anti-VE-cadherin, to substantiate the conclusion that the “lines” observed in the FM1-43 and tomato lectin stainings represent cell-cell interfaces known to contain junctional proteins.

In the figures of the revised manuscript, we still show the “area” of the ECs and not the volume as requested by the reviewer. There are two reasons why we believe this quantification should be preferred over a volume measurement. Firstly, the surface area of ECs defines the area relevant for the diffusion of substances across the cell as well as the density of transporters (together with their absolute numbers), which is the crucial parameter defining the transport rate of an EC in the barrier function of the BBB. The volume plays a less critical role in defining transport at the BBB and largely determines how fast certain levels of molecules are reached within ECs. Secondly, an accurate estimate of the volume of an EC is difficult because its thickness (orthogonal to the lumen) varies considerably from the central region (containing the nucleus) to the tapered peripheral part. We agree with the reviewer that although electron

microscopy does provide a higher spatial resolution, it suffers from the fact that tissue preparation and dehydration lead to significant but not well-defined, tissue shrinkage. Therefore, we do not feel that EM would be optimally suited to address this question.

In the revised manuscript, we present further evidence that FM1-43 does not reach the abluminal membrane of ECs primarily via endocytosis/transcytosis. Live labelling of ECs via the vascular system with tomato lectin strongly stained their luminal membranes but not at all the abluminal membranes (Fig. 3h). If transcytosis would bring FM1-43 molecules that are integrated into the luminal membrane to the abluminal membrane, tomato lectin bound to glycoproteins in the luminal membrane should also accumulate in the abluminal membrane, which we did not observe. We performed additional experiments with FM1-43, in which we increased the concentration of FM1-43 by 10-fold. In those experiments, the abluminal membrane was labelled very quickly and the staining almost reached steady-state levels within 5 min. This rapid labelling is likely driven by a high diffusional flux due to the large concentration gradient of the high FM1-43 concentration in the vessel and luminal membrane towards the abluminal membrane. If this rapid equilibration of fluorescence of the luminal and abluminal membranes would be achieved by transcytosis, then the transcytosis would have to be so fast that an almost complete turnover of labelled and unlabeled membrane on both sides of ECs is possible within only 5 min. This seems unlikely, as the ECs within the BBB exhibit a very suppressed level of transcytosis². In addition, we could not observe FM1-43 labelled transcytotic vesicles in the lumen of ECs even though this dye has already rendered these structures clearly visible in light microscopy³. We therefore suggest that FM1-43 most likely reached the abluminal membrane by diffusion along the membrane and by bypassing junctional proteins at the cell-cell interfaces.

The membrane labelling by FM1-43 is reversible and immediately disappears after removal of the dye from the aqueous solution. Because of this rapid and dynamic membrane labeling FM1-43 is routinely used in synaptic release assays⁴. We have not directly tested the time course of de-staining experimentally, but it is expected that the staining will almost completely disappear within ~15 min. In the revised manuscript we mention the expected reversibility of the staining in the results text for clarity.

3. The authors suggest that FM1-43 may be useful for delivering molecules in the brain by passive diffusion in the cell membrane. Is FM1-43 detected in neurons or astrocytes after a certain time following intra-luminal injection?

The reviewer has a valid point with the suggestion that some accumulation of FM1-43 should be observed in neighboring structures as the dye accumulates in and then leaves the abluminal membrane. Our experiments however did not reveal such an

accumulation. The extent of accumulation could be small and difficult to detect under our experimental conditions. To investigate this possibility experimentally and to increase the accumulation of FM1-43 in the abluminal membrane, we performed additional experiments in which we micro-perfused FM1-43 into the vasculature at a 10-fold higher concentration (Fig. 3f,g). This high concentration rapidly labelled the abluminal membrane (within ~5 min) and indeed resulted in a faint staining of the parenchyma around the vascular system (Suppl Fig. 2b,c). This staining, although weak, was significantly stronger than the staining of an analogous region quantified after tomato lectin perfusion, in which the abluminal membrane was not labelled. It was also not observed at the standard FM1-43 concentration, implying a concentration dependence. The fluorescence of structures in the immediate vicinity of vessels is expected to remain weak, as the dye concentration in the extracellular space of the parenchyma drops steeply with distance – similar to what we observed for the extravasation of several tracers (Fig. 2) – especially as FM1-43 diffuses from a 2D membrane surface. Therefore, adjacent membranes are exposed to a much lower concentration of FM1-43 compared to ECs (lumen contains a high concentration of FM1-43) and are therefore labelled less strongly.

Taken together, we conclude that FM1-43 indeed diffuses out of the abluminal membrane and reaches neighboring structures but produces only a weak fluorescent signal within the time frame of our experiments.

Whether the FM1-43 molecule by itself can act as a potential carrier of other molecules across the BBB is unclear. We wanted to make the point that it is worth considering the route taken by FM1-43 as a new option for overcoming the BBB for drug delivery. To avoid this misunderstanding, we have rephrased this sentence in the revised manuscript accordingly.

4. Rhodamine123 (used in Fig. 3) is a small dye (about 380KDa) that is extruded from the BBB by efflux transporters. This effect is shown in the model after inhibition of the ABCB1 transporter. However, it is surprising that the authors do not find any extravasation of the dye in the brain after inhibition of efflux transport. Please explain.

Indeed, we did not observe rhodamine 123 (Rh123) staining of mitochondria in the brain parenchyma in the presence of verapamil – any fluorescence detected in the brain was so weak that it could not be attributed with certainty to Rh123. The fact that labelling of mitochondria was undetectable in brain parenchyma may be explained by the residual activity of transporters such as ABCC1 (MRP1) and ABCG2 (BRCP)^{5,6}, which are not blocked by verapamil. Both transporters bind to and remove Rh123 such that even in the presence of verapamil cytoplasmic levels of Rh123 in ECs remain low. Because Rh123 has a very high affinity for mitochondria, it accumulates in these

structures and stains them strongly even at the low cytoplasmic concentration observed in ECs. However, under our experimental conditions the low cytoplasmic concentration of RH123 in ECs in combination with the dilution of the dye in the adjacent brain tissue is not sufficient to label mitochondria in adjacent cells in a detectable manner. We have added this explanation to the figure legend.

5. The images in Fig. 3 with the injection of Calcein after ABC transporter inhibition seem to indicate that pericytes could also take up the dye. The authors should evaluate how much of the fluorescence signal corresponds to pericytes or endothelial cells.

We had identified labelled cells as ECs based on their morphology, as they appear to be flat and do not protrude from the vessel outline. To substantiate this conclusion, we performed experiments in which we visualized pericytes with a pericyte-specific live cell tracer (Neurotrace 500/525) and showed that calcein did not accumulate in them (here employing calcein red-orange-AM for spectral separation, Fig. 4h).

6. The authors claim that the effects of DMSO on BBB disruption are mediated by the weakening of the tight junctions. This conclusion is not accurate and not based on data. Can the authors show that the tight junctions are in fact defective? Or that transcytosis is not increased? Is DMSO causing certain toxicity to the endothelial cells?

We agree that we have not provided direct evidence of tight junction damage by DMSO and that the mechanism of action of DMSO remained unresolved. To support the idea that DMSO affects junctional integrity, we have now performed additional experiments in which we injected hyperosmolar mannitol (Fig. 6). This treatment is more clinically established and has been shown to temporarily open the tight junctions within the BBB^{1,7}. Therefore, we expected it to produce a result similar to DMSO if the two treatments share the same mechanism of action. Indeed, we found that TMR extravasation as well as the accumulation of albumin (BSA-Alexa488) in F4/80 positive cells previously observed under DMSO treatment were mimicked by mannitol perfusion (Fig. 6). This similarity of results does not, of course, preclude DMSO from causing them via a different mode of action or that DMSO has additional effects. However, the main purpose of our DMSO experiments was not to reveal the exact mechanism of action of DMSO, but rather to demonstrate that our approach is generally useful for detecting and studying damage to the BBB. We now illustrate this further by applying our approach to the mannitol model and by showing time-lapse imaging of cells taking up albumin (Fig. 6c).

We feel that resolving the exact and complete mechanism of how DMSO damages the BBB is difficult per se as, for example, structural alterations might not necessarily go along with functional changes. Furthermore, BBB leakage may occur in the absence of abnormalities that can be detected by light microscopy. We therefore feel that

addressing this issue would be beyond the scope of this study.

7. The statements about phagocytosis mediated by perivascular macrophages are too strong and not fully substantiated. First, the distinction between microglia and macrophages is more complex and requires more careful classification, i.e. F4/80 is not an exclusive marker for macrophages. Second, the distribution of perivascular macrophages could be different depending on the type of vessels, did the authors observe differences depending on vessel type? Third, the authors say that perivascular macrophages are recruited after DMSO injection, but the "recruitment" mechanism and their "migration" are not explained.

The reviewer raises important questions which had not been fully addressed in the previous version of the manuscript.

In our hands, F4/80 did not stain any cells in the control brain slices even though microglia were present there, as shown by TMEM119 labelling (Fig. 5d). Thus, F4/80 did not stain resting microglial cells and we relied on the literature that F4/80 positive cells in the brain represent macrophages⁸. However, we agree that the definitive identification of perivascular macrophages based solely on the marker F4/80 is difficult and F4/80 positive cells could also represent activated microglia⁹. Therefore, we have revised and weakened this statement.

F4/80 positive cells containing clusters of BSA-Alexa488 were also observed after treating the vascular system with mannitol injections. They represent a consistent phenomenon for both our junctional damage models. F4/80 cells were observed surrounding vessels at a ~2-fold higher frequency around venules over capillaries as we now show in a new analysis (Fig. 6d). The region in which we found F4/80 cells fits the definition of the perivascular space^{10,11} so we now refer to them as F4/80 positive phagocytic cells in the perivascular space and state that they might be perivascular macrophages. We refer to the literature where relevant lineage relationships and marker expression profiles are discussed.

8. The introduction of the epileptic focus is abrupt and not well integrated into the main text. In addition, the results do not appear to be fully consistent, as the authors state that dye extravasation was observed in only one of the animals. The results in human samples also showed no extravasation, but they were performed in patients who had recovered from seizures. Despite the negative results, the authors conclude that they could be caused by a slight weakening of the tight junctions. The authors should be cautious about speculating unless they can show solid evidence to support their hypothesis.

We chose to study BBB alterations in epileptic hippocampi as breakdown as well as increased extrusion capacity have been reported for many years in animal models and in human epilepsy^{12,13}. However, the cellular nature of such changes is not yet fully understood. We subjected epileptic tissue to our novel approach as an example for the application of the technique to a biological question: The function of the BBB in epileptic

tissue.

The unique availability of human biopsies from epileptic patients in our department allowed us to further show that our model is equally well applicable to human tissue.

In the revised manuscript, we added a new set of experiments in which we have also studied the functionality of ABC transporters early after SE. We show that the function of ABCB1/ABCG2 (P-gp/BRCP) is upregulated by ~2-fold such that the extrusion of calcein is more efficient in epileptic mice 2 h post SE than in control mice (Fig. 7g,h). Furthermore, using quantitative mass spectrometry we found that at this early time point after SE, protein levels of ABC transporters are not changed significantly. This implies that the observed early changes must be due to the regulation of transporter function not of protein levels (Fig. 7i).

Furthermore, there seems to be a misunderstanding here due to our unclear description in the initial version of the manuscript: The small leakage of TMR into the parenchyma was observed in every slice of all pilocarpine injected mice. Based on this observation we conclude that there is very limited leakage from the BBB. In contrast, extravasation of BSA-Alexa488 was only weakly observed in one slice, but neither in the other slices nor in the human specimen. The effect in this one slice was clearly not an artifact or background noise. Therefore, as we had no scientific reason to exclude this experiment, we believe it is best to report this observation. To better illustrate this special experimental scatter (“number of cells per volume”) we now show the individual data points.

Minor points:

1. Some of the Figure fields are not described in the main text.

We now refer to every figure panel in the results section.

2. In Fig.3, it would make more sense to show the cartoon of the results according to the experimental data.

The cartoon illustration (Fig. 3a) shows that FM1-43 becomes strongly fluorescent only when integrated into the phospholipid layer constituting the cell membrane, transitioning from the luminal to the abluminal walls, without crossing the phospholipid bilayer, which is consistent with the experimental results obtained with our reported model (Fig. 3d,f,g).

3. The use of diazepam in Fig. 5 is not explained in the main text.

Diazepam (4 mg/kg) was injected to stop the seizures 40 min after their induction. This is necessary for comparability of results between mice but also to minimize the suffering

of the experimental animal to a minimally required degree. We have included this information in the revised results section (legend).

Reviewer #2 (Remarks to the Author):

In this methods paper, Steinlein et al. describe establishing an in situ microperfusion method in brain slices that allows assessing blood-brain barrier function, specifically transporter function and barrier leakage. The idea of such a microperfusion is not new and has been discussed in the brain barriers field for decades.

We are not aware of the fact that discussions about an *in situ* perfusion approach of small vessels in acute brain sections by micro-impalements have been published before but would be happy to include that information and cite relevant papers. Therefore, we would be grateful if some references could be provided.

To the authors' credit, they now established this method. The authors describe perfusing microvessels in mouse brain slices using a set of fluorescent dyes to visualize the vessel lumen and membranes with 2P microscopy.

We would like to thank the reviewer for recognizing that our manuscript represents the first report of an *in situ* microperfusion method in brain slices that allows assessing blood-brain barrier function.

They also used fluorescent efflux transporter substrates and transporter inhibitors to visualize transport activity. In another set of experiments, the authors demonstrate that their method can be used to visualize barrier leakage. Then, the authors used brain slices from mice after status epilepticus and from patients who underwent epilepsy resection surgery and observed mild leakage. Lastly, the authors combined microperfusion with neuroelectrophysiological measurements showing this approach can be used to measure pharmacological effects.

Comments:

This MS is reasonably well-written, but English grammar, language and style need to be revised. For example, the authors use many terms that are simply not used in the English language. Suggest having a native English speaker carefully revise the MS.

We have now revised the manuscript with regard to English grammar and language according to the suggestions of a native US English speaker. If further terms “not used in the English language” remained in the text, we would be grateful for examples so that we can address them specifically.

Given that the authors describe that they could only puncture larger vessels, but not capillaries (lines 162-165), the MS title seems inappropriate.

It is correct that we mostly impale venules in order to micro-perfuse capillaries – we have therefore adapted the title accordingly.

This methods paper provides an interesting new technique to study the blood-brain barrier, but its significance is overstated. Due to the relatively complex setup, it is unlikely this method will be used much. In addition, many researchers in the field are currently using in vivo 2P imaging that accomplishes the same (minus electrophysiology) in a more realistic in vivo setup but without the technological challenges shown here. This should be addressed in the Discussion.

We agree that 2P imaging of the BBB *in vivo* is very powerful and has some advantages over the approach presented here, such as the maintenance of circulating BBB-active factors during the experiment and the possibility of longitudinal imaging to monitor chronic changes in successive recording session. In the revised manuscript, we have tried to provide a balanced discussion of the advantages and disadvantages of our slice approach compared to existing *in vivo* imaging approaches. However, there are significant shortcomings in *in vivo* BBB imaging experiments, as some experiments cannot be performed and some questions cannot be addressed. In contrast to our slice approach, there is little control over the composition of the solution in the vessel (oxygenated blood with blood cells and plasma proteins are required to keep the animal alive) or in the brain parenchyma (it is not possible to immerse the whole brain in an experimental buffer). It is also difficult to achieve a constant concentration of a test molecule in the vessel (due to peripheral excretion and metabolism as well as toxicity *in vivo*). In addition, the choice of molecules is limited by their biocompatibility and the route of entry, which cannot be isolated (entry via brain capillary versus choroid capillaries versus potential glio-lymphatic routes). Finally, it is difficult to reach deeper brain structures – such as the hippocampus – *in vivo* with good subcellular optical resolution.

2P *in vivo* imaging is also used in our lab, in fact on the same microscopes. In our experience, the *in vivo* experiments are significantly more complex and time-consuming, requiring window surgery, anesthesia, more sophisticated imaging tweaks to obtain high-resolution recordings (e.g. syncing with breathing/heart rate, application of a series of image alignment steps) and the throughput is ~3-fold smaller in terms of successful experiments/injections per day. Slice electrophysiology (patch clamp, synaptic potentials, imaging) is a standard technique in many labs. The technical requirements for our slice BBB imaging approach are even lower compared to standard slice electrophysiology, as no electrical recording or stimulation is required. Certainly, it is pure speculation at the moment, but we feel that this new BBB slice model has the potential to become a standard BBB approach in labs equipped for animal experimentation.

The authors criticize existing methods in the field, including in vitro cell culture, isolated

capillaries, and in vivo 2P imaging approaches to assess barrier function, but then only briefly brush over the shortcomings of their model. For example, while the authors correctly point out how in vitro BBB models do not reflect the in vivo situation, they seem to forget that their own model is not an in vivo model but is based on brain slices using artificial buffers, which does not reflect the in vivo situation. In this regard, the authors overinterpret some of their findings and inappropriately overstate the relevance of their approach while at the same time making invalid comparisons with existing models and approaches – these are apple-orange comparisons that are no valid because these models are used for different purposes. The authors should also not forget that what model is used is based on what question is asked. Lastly, other models have been quite useful over many decades but, just like the approach presented here, they are models and at the end of the day: all models are wrong, but some are useful (George Box) – and this certainly includes the model presented here.

We fully agree with the reviewer's view. We are sorry if our discussion of the various models appeared unbalanced and seemed to ignore the weaknesses of the approach. Therefore, we have now revised the relevant discussion section, taking care to point out the limitations of our approach and the type of questions that can be best addressed with this model compared to others.

The M&M section lacks significant experimental detail to properly assess some of what is reported. For example, to improve this MS, the authors should provide pictures of the setup, provide detailed imaging settings, provide details of the SE inductions in mice. Were mice really dosed s.c. or i.p. with methyl-scopolamine, pilocarpine, diazepam? (i.p. is what is mostly used in the literature), provide a video of the mice seizing, provide antibody concentrations (not dilutions).

To address the reviewer's concern we have carefully revised our M&M section. We agree with the reviewer that it would be helpful to add pictures of the recording setup. Therefore, we have included respective pictures in supplementary figure 1 that show the "simplicity" of the setup and its analogy to electrophysiological recording setups. We agree with the criticism that our description of the pilocarpine model was incomplete and inaccurate in some details (imprecise labelling of figure "scopolamine"). The pilocarpine model has been used in the lab for many years and we now refer in the manuscript to some of our papers in which we characterize and describe the model in more detail. We actually inject the drugs mentioned subcutaneously (s.c.) rather than intraperitoneally (i.p.). In our hands, the success rate and reproducibility of the model is better with s.c. than with i.p. injections. Pilocarpine-induced SE was clearly verified visually for each mouse by an experienced investigator and only mice with SE were included in the experiment. Pilocarpine-induced SE can be distinctly recognized and classified and is characterized by lack of movement, increased permanent whisker movements, head tremor and specific tail movement patterns. The onset of SE is usually a convulsive focal to bilateral tonic-clonic seizure. Video documentation is usually not performed as it does not yield any additional information.

Major concerns:

- Figure 5 shows scopolamine – are the authors sure they used methyl-scopolamine and not scopolamine? Even though chemically, the difference is only a methyl-group, this would have significant consequences for the entire study. Specifically, scopolamine is a cholinergic antagonist that, amongst others, is used to generate a chemically-induced dementia model and this effect is based on scopolamine's ability to cross the BBB. The methyl-group in methyl-scopolamine, on the other hand, results in a quaternary ammonium group, resulting in a positively charged molecule that does not cross the BBB. And thus, in the pilocarpine SE model, methyl-scopolamine is used to block peripheral cholinergic effects of pilocarpine. The authors need to unequivocally clarify that methyl-scopolamine was used and not scopolamine. The fact that the mistake in Figure 5 was not recognized by any of the authors prior to the submission of this MS raises serious concerns.

We injected all pilocarpine mice with methyl-scopolamine nitrate (s.c.). We had shortened “methyl-scopolamine” to “scopolamine” for the purpose of labelling the cartoon. The full and correct chemical description was given in the methods section (former page 31). However, we agree that the labeling of the cartoon was misleading with the abbreviation and have changed the labelling within the figure to “methyl-scopolamine” (Fig. 7a).

- Did the authors keep bubbling the aCSF with O₂/CO₂ during the perfusions? If not, have you measured aCSF pH? If it is not bubbled constantly with CO₂, the pH increases to ~10 within minutes after stopping CO₂ bubbling. This would have a dramatic impact on the experiments/data.

Yes, perfusion solutions were constantly aerated with a mixture of O₂/CO₂ (95%/5%) and the pH was verified before each experiment to ensure sufficient pre-bubbling to reach the equilibrium pH of 7.4 (at RT) (Suppl. Fig. 1). During perfusion, the beakers were partially covered to improve gassing of the solution. This is also our standard procedure for slice electrophysiology in the lab. We have additionally clarified this in the methods section.

- The inhibitor concentrations used are clinically unrealistic: clinical verapamil blood levels in patients are in the range 0.2-1 microM. Yet, here the authors used 200 microM. What happens with a clinically realistic verapamil concentration? The authors should provide this data. The same is true for probenecid – 1mM is not realistic clinically – and caffeine (300 microM – show data with a realistic caffeine blood concentration).

The reviewer correctly points to the fact that the clinically used concentrations of these drugs may indeed be much lower. Therefore, we would like to explain the strategy behind our choice of concentrations for these drugs. The purpose of applying verapamil and probenecid was not to mimic or investigate the effects of these drugs in vivo or in humans. Rather, we used them as established experimental, pharmacological inhibitors of some ABC transporters. The concentrations we applied (200 µM and 1 mM

for verapamil and probenecid, respectively) are in the upper range of concentrations used previously for this purpose¹⁴⁻¹⁸. We chose the upper end of the range as we aimed for an almost complete blockade of the ABC transporters bound by these drugs to isolate the effect of the unblocked transporters.

The required concentration of caffeine as an antagonist at A1 receptors is determined by the concentration and affinity of the agonist used to reduce fEPSPs. We used 15 nM N6-cyclopentyladenosine (CPA) to activate inhibitory A1 receptors, which is ~6-fold higher than the agonist's affinity at this receptor and produces an almost maximal degree of A1 receptor activation. As we demonstrated in the former Fig. 6a (now Fig. 8a), 300 μ M was sufficiently potent to antagonize A1 receptor activation. This concentration agrees well with the published pharmacology of A1 receptors: ~100 μ M caffeine is required for a half maximal reduction of 1 nM CPA binding to A1 receptors (affinity ~1 nM at A1 receptors). As we apply CPA strongly above its affinity, it is expected that we also need to increase the concentration of the antagonist caffeine. We realized that the concentration of CPA employed in our experiments was not stated in the results section (only methods) and have now included this information.

- *Rhodamine123 is also not a terribly good Pgp substrate, the authors should have used something else.*

Under control conditions, Rh123 did not accumulate in ECs to a detectable degree and did not stain their mitochondria despite its good membrane permeability. This means that it was a sufficiently good substrate for extrusion mechanisms in ECs. Blocking P-gp transporters with verapamil resulted in an accumulation of Rh123 in ECs and a strong staining of their mitochondria, showing that P-gp transporters substantially contribute to the extrusion of Rh123 under control conditions. This result should demonstrate that our approach, using this or other ABC transporter substrates, allows to perform functional assays on the cellular level in the native BBB. There may be better P-gp substrates that could be used, but we do not see how these substrates would strengthen the conclusion from this experiment.

- *Group numbers for the mouse experiments (n=2-3) are too low.*

For all experiments we collected data from 4-9 mice with a few exceptions:

- Microglial and macrophage staining of slices previously treated with DMSO/control. Here we used 3/3/3/2 mice for the 4 different groups and 7/4/7/3 slices per group.

We now added 4 mice and 12 slices that were subjected to mannitol treatment and successive immunostaining.

- In the pilocarpine experiments, we collected data from 3 mice with 14 slice recordings at 2 h post SE.

It is important to emphasize that in cellular neurobiology the number of investigated cells (e.g. patch-clamped) or the number of recorded slices is usually considered the relevant measure of the number of biologically independent replicates. In this study, each slice is treated separately with an impalement and is perfused with a separately prepared solution and so forth. The variability of results from slice to slice/injection to injection is considerably larger than that from mouse to mouse. This is due to the many manipulations required to achieve the individual recording (pipette positioning, selection of vessel, quality of impalement, etc.). This is very different from behavioral experiments or in vivo imaging recordings, where there is only one “preparation” per mouse and the number of replicates is usually based on the number of mice investigated.

Thus, the only group in which the n count is less than 4 is DMSO injection + macrophage staining F4/80 – and for this question we have now added 4 slices treated with mannitol to open tight junctions and successively stained for F4/80, confirming the results (BSA-Alexa488 in F4/80 positive cells) obtained with 3 DMSO-treated/injected slices.

Overall, we feel that our experimental groups are sufficiently large to draw firm conclusions.

- *Provide more details of how fluorescence was measured to obtain Figs. 1, 2.*

We have included detailed descriptions of how the fluorescence image-based parameters were calculated for all panels of figures 1 and 2 in the methods section.

The Introduction, Results, and Discussion sections need to be revised as they suggest a significant lack of knowledge in the blood-brain barrier field. Examples:

- *Lines 51-53: The NVU traditionally includes neurons.*

We have corrected this error.

- *Lines 69-77: A lot of work has been done over the last 20 years that answers these questions (at least in part). Yet, none of this work has been cited. Recommend you read up on the works of the following groups: Betsholtz, Engelhardt, Miller/Bauer/Hartz, Ronaldson/Davis, Renier, Erturk and others.*

We agree that the way we wrote this passage (without citations) was somewhat unfortunate, as it may have generated the impression that the listed questions have not been addressed so far. Our intention was not to neglect or ignore relevant studies, but

rather to point out that addressing the listed important open questions (from expert reviews) would be facilitated if novel approaches to study the BBB became available. Certainly, we cannot provide a complete review of the field, as we have already cited 70 papers, but we have now cited several highlight papers of the groups mentioned in order to indicate that significant progress has been made and hope that thereby we now achieved of a more balanced presentation of the topic.

- Lines 88-90: The authors seem to be unaware of Kariolis et al. 2020 and Ullman et al. 2020

We were indeed not aware of those studies and are grateful for these references. We now cite these papers and have reworded our statement in the introduction to reflect the clear progress of the two studies in the context of using transcytosis to overcome the BBB.

- Lines 90-93: Existing literature shows that an open barrier does NOT lead to better drug permeation. Epilepsy and brain cancer are good examples: while the barrier is leaky in some areas, overexpression of transporters seems to compensate that (see for the most recent publication deGooijer et al. 2021).

We think that the reviewer refers to the following phrase “thereby potentially leading to a better drug permeation”. We deleted this part and reworded the sentence. Our main point was that the assessment of BBB function in inflammatory diseases could be improved with approaches which allow direct visualization and identification of the cell layer acting as a barrier, in this example/cited work, a layer of astrocytes formed a barrier under inflammatory conditions¹⁹.

- Lines 94-99: These are all good points but the authors’ model has similar weaknesses that are not mentioned.

We have now extended the discussion of our model and mentioned more explicitly the weaknesses of our model (lines 506-516). We believe that the points raised in lines 94-99 are well covered by our approach. The slice contains a “native BBB”, which has been in interaction with circulating molecules (prior to slice preparation, in contrast to in vitro approaches) and provides good control of the intra-luminal solution. But we agree that our model is not the all fits one solution of BBB research.

94 With the currently available methodology, answering these open questions is rather difficult.
95 Approaches are required that allow to analyze the native BBB, which has been in interaction with
96 the healthy or the diseased brain and was under the control of systemically circulating signaling
97 molecules. Furthermore, experimentally controlling the intra-luminal solution is of paramount
98 importance in order to apply or interfere with BBB-activating molecules, cells, drug candidates or
99 fluorescent tracers. Suitable experimental approaches should be applicable to various brain

- Lines 198-199: This is nothing new - it is known that tight junctions limited diffusion within the membrane. In fact, it is the tight junctions that separate the luminal vs. abluminal membranes, thus creating the polarization of the brain endothelium.

We agree that the fact that tight junctions limit membrane diffusion is well recognized in the field. However, to our knowledge, the data showing that tight junctions represent a “fence” in the membrane have largely been obtained in epithelial cells and not in brain capillaries. In our view, the FM1-43 experiments presented here are the first experimental data for a fence function of junctional proteins at the BBB.

Moreover, our experiments show that – contrary to previous assumptions – the fence function is relative, and that FM1-43 can indeed reach the abluminal membrane. We substantiate this finding in the revised manuscript by increasing the FM1-43 concentration in new experiments and by showing that under these conditions the abluminal membrane is completely stained within 5 min (Fig. 3e,f). We feel that this is a clear and important contribution to the current state of research, but are happy to revise this claim if we have overlooked previous key publications on a fencing function of tight junctions at the BBB.

- Lines 251-255: This conclusion (Mrp1 is not functional) is not necessarily correct: P-gp levels are much higher than those of Mrp1 and blocking Pgp could simply overwhelm lower Mrp1 levels, and thus, result in staining.

We fully agree with the reviewer that the statement that “MRP1 is not functional” is too strong. What we actually meant to say was that the contribution of MRP1 to calcein removal is much smaller than that of P-gp/BRCP based on the facts that a) MRP1 alone cannot entirely prevent calcein accumulation and b) blocking MRP1 reduced the overall transport rate by an unnoticeable extent, so that no accumulation of the dye was observed. We have rephrased our conclusion to explicitly state that MRP1 may well be functional albeit at a lower level than P-gp when tested with calcein-AM.

Figure 2D: at the bottom of that microvessel (center of image) there seems to be a cell. Is that a pericyte? If yes, then what is observed is not the abluminal membrane (lines 196-198).

The cell referred to by the reviewer is an EC – we identify it as such based on its morphology. This is difficult to see in the small overlay image alone but becomes clear when the channels are viewed separately and by comparing the cell's morphology to that of a pericyte in the Dotd scan mode (now depicted in Suppl Fig. 2a,d). There is almost no “tissue material” between the EC and the lumen and the cell appears flat and elongated. In contrast, pericytes have a much rounder soma that protrudes from the vessel wall. This morphological classification was confirmed by labelling with calcein red-orange-AM and Neurotrace 500/525.

Line 280: using 10% DMSO is a reasonable tool, but clinically unrealistic. Why not use 25% mannitol (as is used in the clinic) to osmotically open the barrier (this is used at OHSU, by the Neuwelt group to open the barrier to deliver chemotherapeutics to brain cancer patients)?

We followed the reviewer's suggestion and performed a new series of experiments in which we perfused the vessels with 15% mannitol to open the BBB. We observed a similar extent of TMR leakage and extravasation of BSA-Alexa488 in vessels treated with 15% mannitol compared to those treated with DMSO, supporting our conclusion that DMSO opens the BBB in a similar manner to mannitol^{1,7}. These new experiments are now shown in Figure 6.

Line 281: BSA-Alexa488 – did the authors purify this product before using it? The reason for purifying BSA-Alexa488 (and similar products) is that there is often unconjugated Alexa488 in these products leading to false data and misinterpretations because observations are based on the dye alone and not the conjugated product due to impurities.

Purified BSA-Alexa488 was purchased from Thermofisher (cat# A13100) and kept frozen in aliquots before use. We also tested unconjugated Alexa488 as a soluble tracer (not shown) which does not cross the BBB (no leakage or extravasation in the form of accumulations or clusters). Furthermore, even after opening the BBB with either DMSO or mannitol, traces of the dye leaking into the parenchyma stained the tissue diffusely but did not accumulate in fluorescent granules (TMR, similar to Alexa488). This was only observed when applying the BSA-Alexa488 conjugate. Overall, we are confident that this phenomenon is not caused by a potential contamination of the BSA-Alexa488 solution with unconjugated Alexa488 but depends on the presence of BSA which becomes visible through the dye bound to it.

Line 302: DMSO-injured TJs – that's a speculation you have not proof for. DMSO increases membrane fluidity. Is it possible that this may have lead to this observation?

It is correct that we have no direct evidence that DMSO damages tight junctions. This

was an indirect conclusion by exclusion based on the fact that we found BSA-Alexa488 (in macrophages/microglial cells) and TMR (in the interstitium) to cross the BBB and that we could not detect signs of transcytosis for either molecule (no fluorescent puncta in the cytoplasm of ECs as observed in³). To strengthen our conclusion of tight junction damage, we repeated these experiments with mannitol, a clinically established maneuver to open tight junctions^{1,7}, and now report that it reproduces the effect observed after DMSO application (Fig. 6, TMR leakage, extravasation of BSA-Alexa488, recruitment of F4/80 positive cells).

It still remains possible that DMSO has other or additional effects that can cause the same changes to a similar extent via other mechanisms. As mentioned by the reviewer, DMSO increases membrane fluidity and can cause transient pores in the membrane²⁰. On one hand, we have never observed TMR or BSA-Alexa488 in the cytoplasm of ECs, making the presence of pores unlikely. On the other hand, DMSO is also known to impact protein-protein interactions and protein conformation²¹⁻²³. This effect may explain how DMSO could impact junctional protein function in the BBB.

However, the main purpose of our DMSO experiments was to illustrate the applicability of our model to study disturbances of the BBB at the subcellular level in real time. In this context, DMSO was considered a “tool to damage the BBB” and we believe that exploring the many aspects of the effect of DMSO on the BBB (concentration and duration of application, targeted cell type etc.) is beyond the scope of our study.

Line 338: SE-triggered changes to the TJs – that’s again a speculation that is not backed up by data. While it is true that SE induces loss of TJs, these effects last more than 24h.

We agree that we cannot demonstrate direct damage to tight junctions after SE and have therefore now weakened this conclusion accordingly in the revised manuscript. We have based this interpretation on the similarity of observed alterations compared to the control condition between post-SE slices and slices treated with either DMSO or mannitol with regard to leakage of tracer into the parenchyma and the absence of dye within ECs. In our view, this is most easily explained by paracellular leakage across tight junctions, but other mechanisms may be possible as well.

Current understanding in the field is that the two most important BBB efflux transporters are Pgp and BCRP – not Mrp1. Why did the authors look at Mrp1 and not BCRP? This seems like a missed opportunity (or another lack of knowledge of the field?) and the authors should provide data showing BCRP-mediated transport.

It is true that none of our assays tests the functionality of BCRP1 and it may be seen as a missed opportunity as along with the right substrate/blocker combination our model is ideally suited for such a test.

On the other hand, we would like to point out that the purpose of the manuscript is to show the broad applicability of the approach of intravascular micro-perfusion (permeability, EC structure/size, membrane diffusion, ABC activity calcein-AM/rhodamine123, chemical and physical damage and protein extravasation, epilepsy, human biopsy material, electrophysiology). This implies that not every individual application can be performed in an almost complete manner, i.e. that some opportunities have to be missed.

The authors should add data from experiments assessing Pgp-mediated transport in brain slices from SE mice and/or epilepsy patients. Efflux transporters are upregulated after seizures and this is something the authors should be able to detect with their system.

We are grateful for this suggestion and have now performed calcein-AM uptake assay experiments early after SE. In mouse slices that had experienced SE as well as in mice injected with saline, no accumulation of calcein was observed in ECs demonstrating that there is no gross functional loss of ABCB1 transporters (P-pg). However, when we partially blocked transporters (P-pg/MRP1) with a lower concentration of Elacridar/probenecid to cause some accumulation in the control slices, we observed that the accumulation of calcein in the SE-slices was two-fold or significantly lower (Fig. 7g,h). This result indicates that the calcein extrusion rate of ECs is ~2-fold higher 2 h after SE, most likely due to increased activity of P-pg. Interestingly, our new correlated mass spectrometry analysis early after SE (Fig. 7i) showed no significant upregulation of the protein levels of P-pg (ABCB1), suggesting that the increased transport resulted from a functional regulation on the post-translational level. Although not supported by statistical significance, we found that ABCB1/8 and ABCG2 are the only ABC transporters with a trend towards upregulation early after SE. However, even if those changes were significant, the degree of regulation is too low (<20%) to explain the doubled extrusion rate observed functionally in the uptake assay.

References:

1. Rapoport, S. I. Osmotic Opening of the Blood–Brain Barrier: Principles, Mechanism, and Therapeutic Applications. *Cell Mol Neurobiol* 20, 217–230 (2000).
2. Andreone, B. J. *et al.* Blood-Brain Barrier Permeability Is Regulated by Lipid Transport-Dependent Suppression of Caveolae-Mediated Transcytosis. *Neuron* 94, 581-594.e5 (2017).
3. Raucher, D. & Sheetz, M. P. Membrane Expansion Increases Endocytosis Rate during Mitosis. *J Cell Biology* 144, 497–506 (1999).

4. Kavalali, E. T. & Jorgensen, E. M. Visualizing presynaptic function. *Nat Neurosci* 17, 10–16 (2014).
5. Henrich, C. J. *et al.* A High-Throughput Cell-Based Assay for Inhibitors of ABCG2 Activity. *J Biomol Screen* 11, 176–183 (2006).
6. Saengkhae, C., Loetchutinat, C. & Garnier-Suillerot, A. Kinetic Analysis of Rhodamines Efflux Mediated by the Multidrug Resistance Protein (MRP1). *Biophys J* 85, 2006–2014 (2003).
7. Al-Sarraf, H., Ghaaedi, F. & Redzic, Z. Time Course of Hyperosmolar Opening of the Blood-Brain and Blood-CSF Barriers in Spontaneously Hypertensive Rats. *J Vasc Res* 44, 99–109 (2007).
8. Yang, T., Guo, R. & Zhang, F. Brain perivascular macrophages: Recent advances and implications in health and diseases. *Cns Neurosci Ther* 25, 1318–1328 (2019).
9. Butovsky, O. & Weiner, H. L. Microglial signatures and their role in health and disease. *Nat Rev Neurosci* 19, 622–635 (2018).
10. Owens, T., Bechmann, I. & Engelhardt, B. Perivascular Spaces and the Two Steps to Neuroinflammation. *J Neuropathology Exp Neurol* 67, 1113–1121 (2008).
11. Mastorakos, P. & McGavern, D. The anatomy and immunology of vasculature in the central nervous system. *Sci Immunol* 4, (2019).
12. Vliet, E. A. van, Aronica, E. & Gorter, J. A. Blood–brain barrier dysfunction, seizures and epilepsy. *Semin Cell Dev Biol* 38, 26–34 (2015).
13. Löscher, W. & Friedman, A. Structural, Molecular, and Functional Alterations of the Blood-Brain Barrier during Epileptogenesis and Epilepsy: A Cause, Consequence, or Both? *Int J Mol Sci* 21, 591 (2020).
14. Flores, K., Manautou, J. E. & Renfro, J. L. Gender-specific expression of ATP-binding cassette (Abc) transporters and cytoprotective genes in mouse choroid plexus. *Toxicology* 386, 84–92 (2017).
15. Gollapudi, S., Kim, C. H., Tran, B.-N., Sangha, S. & Gupta, S. Probenecid reverses multidrug resistance in multidrug resistance-associated protein-overexpressing HL60/AR and H69/AR cells but not in P-glycoprotein-overexpressing HL60/Tax and P388/ADR cells. *Cancer Chemoth Pharm* 40, 150–158 (1997).
16. Cisternino, S., Rousselle, C., Lorico, A., Rappa, G. & Scherrmann, J.-M. Apparent Lack of Mrp1-Mediated Efflux at the Luminal Side of Mouse Blood-Brain Barrier Endothelial Cells. *Pharmaceut Res* 20, 904–909 (2003).
17. Neuhaus, W. *et al.* Blood–brain barrier cell line PBMEC/C1-2 possesses functionally active P-glycoprotein. *Neurosci Lett* 469, 224–228 (2010).
18. Gedeon, C., Behravan, J., Koren, G. & Piquette-Miller, M. Transport of Glyburide by Placental ABC Transporters: Implications in Fetal Drug Exposure. *Placenta* 27, 1096–1102 (2006).

19. Horng, S. *et al.* Astrocytic tight junctions control inflammatory CNS lesion pathogenesis. *J Clin Invest* 127, 3136–3151 (2017).
20. Gurtovenko, A. A. & Anwar, J. Modulating the Structure and Properties of Cell Membranes: The Molecular Mechanism of Action of Dimethyl Sulfoxide. *J Phys Chem B* 111, 10453–10460 (2007).
21. Chan, D. S.-H. *et al.* Effect of DMSO on Protein Structure and Interactions Assessed by Collision-Induced Dissociation and Unfolding. *Anal Chem* 89, 9976–9983 (2017).
22. Tjernberg, A., Markova, N., Griffiths, W. J. & Hallén, D. DMSO-Related Effects in Protein Characterization. *J Biomol Screen* 11, 131–137 (2006).
23. Arakawa, T., Kita, Y. & Timasheff, S. N. Protein precipitation and denaturation by dimethyl sulfoxide. *Biophys Chem* 131, 62–70 (2007).
24. Bruin, M. de, Miyake, K., Litman, T., Robey, R. & Bates, S. E. Reversal of resistance by GF120918 in cell lines expressing the ABC half-transporter, MXR. *Cancer Lett* 146, 117–126 (1999).

REVIEWER COMMENTS

Reviewer #2 (Remarks to the Author):

In this revised methods paper, Hanafy and Steinlein et al. adequately addressed most of this reviewer's comments and questions and, as a result, the MS is much improved. Some questions remain and this reviewer also made a few more comments below that should all be fully addressed. As stated in my previous comments, this is an interesting manuscript that describes a technically challenging study. However, I also maintain that the scientific impact of this study is limited due to a) the highly specialized instrumental and technical requirements that are needed and that will be an obstacle for its wider use in the field and, more importantly, b) the lack of novel scientific findings this method will afford. While the ability of microperfusing larger brain vessels while conducting neuroelectrophysiological measurements is a technological advance, perfusing microvessels/capillaries is not (yet?) possible with this method, which would be more interesting to the blood-brain barrier field. While this microperfusion method is a "cool" technique, it is unclear what new paradigm-shifting scientific discoveries can be made with it that are not possible with currently existing methods and techniques. At the very least, this should be pointed out in the discussion/conclusions.

Comments:

Apologies if I missed this in the revised MS, but clearly state that non-fluorescent calcein-AM is a P-gp substrate and that the cleaved and fluorescent calcein is a Mrp substrate.

The authors observe that P-gp activity is increased 2h after SE but that P-gp protein levels are not increased and speculate that this could be due to "functional regulation on the post-translational level". This is understandable handwaving when one does not know the answer. However, I would contest this explanation: 20-25 years ago, various researchers tested xenobiotics for their interactions with P-gp using the calcein-AM assay in cell cultures in vitro. Shortly after the cells were exposed to xenobiotics, there was a fast (minutes) response indicating an increase in Pgp activity (20-40% increase from baseline). When trafficking inhibitors were added, this effect was gone. The explanation was that upon first exposure to xenobiotics, the cells respond by trafficking subapical Pgp vesicles into the luminal membrane and activating a second defense line ("mobilizing the national guard"). This would be a reasonable explanation for quickly increased Pgp activity levels, yet constant Pgp expression levels. However, this reviewer is not aware of whether any of this has been published or if this remains anecdotal.

Again, apologies if I missed this in the revised MS: for how long can brain slices be maintained in buffer and be used to conduct useful measurements? If this is not mentioned in the MS, it should.

Clarification: based on the misunderstanding re: scopolamine vs methyl-scopolamine, please confirm that sulforhodamine 101 was used (or was it sulforhodamine 101 acid chloride?). Since these are Mrp substrates, would this be a consideration in how some of the data would be interpreted?

How does the lack of perfusion in the brain slice preparation affect barrier properties? Is this possible to test? If not, this should be discussed as one of the disadvantages/unknowns in this system vs. other approaches.

Statistics/group numbers:

If N is 10 or less per group, it is strongly recommended that individual data points be represented on charts (this is done on some, but not all). If N is less than 5 on a given test, the authors should explicitly note this in the text and be cautious in interpretation of significant results.

Statistical terminology is odd: a “dataset” refers to all the data collected from an experiment. The authors should use the term “treatment”, not “dataset”.

Did the authors use unprotected multiple t tests when performing pairwise comparisons after ANOVA when there were more than two treatments? If so, it is recommended they repeat their pairwise comparisons with adjustment for multiple comparisons.

Reviewer #3 (Remarks to the Author):

One of the limitations that have not been discussed is the intrinsic problem of media perfusion, which removes blood-borne differences from the intraluminal perfusate. This is in particular a problem with pilocarpine: when administered peripherally, pilocarpine exerts its pro-seizure effects by a peripheral mechanism¹⁻⁴. This needs to be discussed.

This paper is testimony of technological and manual excellency. I find the approach used extremely appealing and perhaps not too difficult to adopt if someone with an electrophysiological background is available. The results are often very solid, and the overall quality of data is impressive. There are still several odd sentences that require professional editing, but this is easily done. There are methodological and conceptual issues however, which I describe below. In general, they are related to the translational relevance of the paper. I suggest that the Authors consult with those who developed mannitol or BBB disruption (or others who still use it), and to read with more attention the literature on 3D models and pilocarpine SE. In other words, an exceptional paper with a few non-fatal deficiencies.

Line 111: It is not true that 3D models of the BBB lack perfusion! The first model (3D and perfusion) is from the 1990s (Stanness et al., see also 5, 6). Thus, this sentence in the introduction needs to be modified to acknowledge the state of art in the field. Same for statement line 113: The models cited above have been extensively used to study drug delivery issues e.g., 7. A distinct advantage of this approach is exactly what the Authors try to achieve with pilocarpine: study drug passage in epilepsies. We the model cite above, this is done directly on human epileptic brain endothelial cells^{8, 9}. Please discuss.

Did you record any electrical activity in sclerotic hippocampi from brain resection? What part of the hippocampus did you study (DG, CA1, etc.). Shouldn't the sclerotic hippocampus be permeant to

gadolinium? Please show the contrast MRI from the same hippocampus to convince the reader that here was no leakage in vivo, or remove this section from the paper. The “sturdiness” of the vessels may be due to the sclerotic nature of the tissue.

A reviewer noted that surface not volume was reported for the ROI. This is not at all trivial as the rebuttal suggests in particular if drug passage is being measured. In the equation $\text{Permeability} = \text{Papp} * S$, the term Papp contains the volumes of the luminal and abluminal tracer (presented as concentrations). Please address.

There is a great deal of confusion as to the experiments with mannitol. The concentration of use for BBB disruption is not 15%, which is used to reduce brain swelling. The molarity of the solution used by Neuwelt and others in three times the osmolarity of blood, injected intracarotid. The Authors need to explain their findings with a concentration that in fact is used clinically to improve BBB function. What were the effects of mannitol on vessel diameter?

The concentration of probenecid is very high and there is concern on the vehicle effects without the drug. Any issue here worth discussing? What were the effects of vehicle alone? Also, you report 1 mM and 0.6 mM. Which one was used? And if both were used, why the different concentrations?

Question 1: Is there a readout of the pipette perfusion pressure, beginning and end?

Question 2: Would it be possible to add leukocytes to the perfusate? See comment above.

1. Marchi N, Johnson AJ, Puvenna V, Johnson HL, Tierney W, Ghosh C, Cucullo L, Fabene PF, Janigro D. Modulation of peripheral cytotoxic cells and ictogenesis in a model of seizures. *Epilepsia*. 2011;52(9):1627-34. Epub 2011/06/02. doi: 10.1111/j.1528-1167.2011.03080.x. PubMed PMID: 21627645; PMCID: PMC3728674.
2. Uva L, Librizzi L, Marchi N, Noe F, Bongiovanni R, Vezzani A, Janigro D, de Curtis M. Acute induction of epileptiform discharges by pilocarpine in the in vitro isolated guinea-pig brain requires enhancement of blood-brain barrier permeability. *Neuroscience*. 2008;151(1):303-12. Epub 2007/12/18. doi: 10.1016/j.neuroscience.2007.10.037. PubMed PMID: 18082973; PMCID: PMC2774816.
3. Marchi N, Oby E, Fernandez N, Uva L, de Curtis M, Batra A, Santaguida S, Barnes V, van Boxel A, Najm I, Janigro D. In vivo and in vitro effects of pilocarpine: relevance to epileptogenesis. *Epilepsia*. 2007;48(10):1934-46.
4. Uva L, Janigro D, de Curtis M. Submillimolar and supramillimolar concentrations of pilocarpine induce, respectively, gamma oscillations and epileptiform activity in the in vitro isolated guinea pig brain. *Soc for Neuroscience Annual Meeting*. 2006;Atlanta, GA.
5. Cucullo L, Hossain M, Tierney W, Janigro D. A new dynamic in vitro modular capillaries-venules modular system: cerebrovascular physiology in a box. *BMC Neurosci*. 2013;14:18. Epub 2013/02/08. doi: 10.1186/1471-2202-14-18. PubMed PMID: 23388041; PMCID: PMC3598202.
6. Cucullo L, Hossain M, Puvenna V, Marchi N, Janigro D. The role of shear stress in Blood-Brain Barrier endothelial physiology. *BMC Neurosci*. 2011;12:40. Epub 2011/05/17. doi: 10.1186/1471-2202-12-40. PubMed PMID: 21569296; PMCID: PMC3103473.
7. Williams-Medina A, Deblock M, Janigro D. In vitro Models of the Blood-Brain Barrier: Tools in Translational Medicine. *Front Med Technol*. 2020;2:623950. Epub 20210215. doi: 10.3389/fmedt.2020.623950. PubMed PMID: 35047899; PMCID: PMC8757867.
8. Ghosh C, Marchi N, Hossain M, Rasmussen P, Alexopoulos AV, Gonzalez-Martinez J, Yang H, Janigro D.

A pro-convulsive carbamazepine metabolite: quinolinic acid in drug resistant epileptic human brain. *Neurobiol Dis.* 2012;46(3):692-700. Epub 2012/03/20. doi: 10.1016/j.nbd.2012.03.010. PubMed PMID: 22426401; PMCID: PMC4001854.

9. Santaguida S, Janigro D, Hossain M, Oby E, Rapp E, Cucullo L. Side by side comparison between dynamic versus static models of blood-brain barrier in vitro: a permeability study. *Brain Res.* 2006;1109(1):1-13. Epub 2006/07/22. doi: 10.1016/j.brainres.2006.06.027. PubMed PMID: 16857178.

General remarks – corrections of data in this version

There are some small differences in the mass spec results compared to the previous submission. During re-revising the ms we figured out that for the MaxQuant search some parameters had not been correctly set (e.g. phosphorylation was a variable modification, when it should not have been present. Carbamidomethylation was a variable modification when it should have been fixed). The corrected data now is displayed, mentioned in the results and uploaded to PRIDE.

As you can see below (left: old, right: new and corrected) the same ABC transporters were identified and the Pilo-induced change is almost the same and still extremely small. Some significances shifted as they were and still are borderline. Anyway, we concluded and still conclude from this data that there is little if any regulation of ABC transporter levels at 4h and even at 24h it is within $\pm 10\%$ (was between $\pm 12\%$) and this cannot account for the increase in transport activity we have observed.

On inspection of figure 8 c we discovered that some data points had erroneously not been included in the averages (grey and red bars, left old version, right new version). Therefore, the corrected mean values have slightly changed in the present version. However, this does not affect the statistical outcome and also not the biological meaning: in both cases caffeine applied i.v./ via intravascular injection increases field potentials but ACSF i.v. does not. Caffeine directly applied to the bath chamber in both cases has a numerically larger effect, but this is not significantly different.

Reviewer #2 (Remarks to the Author):

In this revised methods paper, Hanafy and Steinlein et al. adequately addressed most of this reviewer's comments and questions and, as a result, the MS is much improved. Some questions remain and this reviewer also made a few more comments below that should all be fully addressed. As stated in my previous comments, this is an interesting manuscript that describes a technically challenging study. However, I also maintain that the scientific

impact of this study is limited due to a) the highly specialized instrumental and technical requirements that are needed and that will be an obstacle for its wider use in the field and, more importantly, b) the lack of novel scientific findings this method will afford.

We appreciate that this reviewer considers the revised manuscript as “much improved” and “interesting”.

We would like to point out that 2-photon imaging in combination with electrophysiological recordings has become a standard technique in many laboratories. No additional instrumentation is required and we share the view of reviewer #3 that it is “not too difficult” to set up this method. Of course, only the future will show how commonly this method is going to be applied. However, we are convinced that this technical approach has the potential to be applied routinely in BBB research. In our lab, we are currently running follow-up projects using this technique to study developmental and Alzheimer related changes of the BBB.

Furthermore, we would like to draw the attention of the reviewer to the following novel scientific findings of our study which have been obtained with this new method:

- Permeation of the native BBB by a very small (200D) polar tracer (7HCC) with a permeability cut-off at ~500-600D (Fig. 2).
- Existence of membrane-delimited diffusion across tight junction complexes (Fig. 3).
- Brain capillaries show a constant ratio of EC size to vessel size of 0.2 (Fig. 3), a universal number to extract and estimate the number of ECs along capillaries.
- Functional assays of ABC transporters showing that the potency to transport calcein-AM, Rh123 and Hoechst is at least an order of magnitude higher than previously estimated *in vitro* (Fig. 4).
- Recruitment of F4/80 positive phagocytotic cells by BBB damage and extravasation of albumin (Fig. 5 & 6).
- DMSO and mannitol produce similar alterations of the BBB (Fig. 5 & 6).
- ABCB1/ABCC1 transporter functionality is increased 2 h after *status epilepticus*, tight junctions slightly open, almost no albumin extravasation (Fig. 7).

While the ability of microperfusing larger brain vessels while conducting neuroelectrophysiological measurements is a technological advance, perfusing microvessels/capillaries is not (yet?) possible with this method, which would be more interesting to the blood-brain barrier field. While this microperfusion method is a “cool” technique, it is unclear what new paradigm-shifting scientific discoveries can be made with it that are not possible with currently existing methods and techniques. At the very least, this

should be pointed out in the discussion/conclusions.

We believe that there might be a misunderstanding: We perform the impalement of the vasculature in larger vessels, typically venules, but we perfuse the whole vasculature including capillaries and perform our analysis at capillaries, the very relevant part of the BBB, as pointed out by the reviewer (for example see Fig. 2). Venules are only our entrance door to the lumen of capillaries. Having this point of impalement spatially separated from the analysis is a technical advantage as it minimizes any contamination of the fluorescent signal around capillaries should a small leakage occur around the impalement site.

We describe in detail the impalement site and strategy page 150ff.

Comments:

Apologies if I missed this in the revised MS, but clearly state that non-fluorescent calcein-AM is a P-gp substrate and that the cleaved and fluorescent calcein is a Mrp substrate.

We are grateful for this comment and now clearly state that calcein-AM is a substrate of ABCB1 (P-gp) and ABCC1 (MRP1), while fluorescent calcein is a substrate of only ABCC1 (MRP1) (lines 268-269).

The authors observe that P-gp activity is increased 2h after SE but that P-gp protein levels are not increased and speculate that this could be due to “functional regulation on the post-translational level”. This is understandable handwaving when one does not know the answer. However, I would contest this explanation: 20-25 years ago, various researchers tested xenobiotics for their interactions with P-gp using the calcein-AM assay in cell cultures in vitro. Shortly after the cells were exposed to xenobiotics, there was a fast (minutes) response indicating an increase in Pgp activity (20-40% increase from baseline). When trafficking inhibitors were added, this effect was gone. The explanation was that upon first exposure to xenobiotics, the cells respond by trafficking subapical Pgp vesicles into the luminal membrane and activating a second defense line (“mobilizing the national guard”). This would be a reasonable explanation for quickly increased Pgp activity levels, yet constant Pgp expression levels. However, this reviewer is not aware of whether any of this has been published or if this remains anecdotal.

In the revised version we had not used the phrase “functional regulation on the post-translational level”. In lines 401-403 we wrote:

“...3-4-fold-increase in calcein-AM extrusion rate of ECs from pilocarpine-treated mice can hardly be explained by increased protein expression but rather is due to a functional regulation.”.

“Functional regulation” in our view could include a change in P-gp trafficking but is admittedly a bit unspecific. Therefore, we thank the reviewer for bringing up the finding of pain-induced P-gp trafficking demonstrated by McCaffrey *et al.*⁽¹⁾ and follow-up studies and now explicitly mention that mechanism in lines 403-404.

Again, apologies if I missed this in the revised MS: for how long can brain slices be maintained in buffer and be used to conduct useful measurements? If this is not mentioned in the MS, it should.

We agree with the reviewer that this is important information. We have sharpened our statement in the methods section (lines 781-782). The acute slices can be reliably used to study the BBB for up to 6-8 h following slicing.

Clarification: based on the misunderstanding re: scopolamine vs methyl-scopolamine, please confirm that sulforhodamine 101 was used (or was it sulforhodamine 101 acid chloride?). Since these are Mrp substrates, would this be a consideration in how some of the data would be interpreted?

In this study, we used sulforhodamine 101 (MW 607, not the chloride salt) as an example of a medium-sized tracer to investigate potential paracellular diffusion through tight junctions (Fig. 2b). In that assay, we treated any fluorescence appearing outside the capillary as being leaked out by paracellular diffusion. This interpretation is justified as long as there is no evidence for transcellular diffusion or diffusion via endothelial cells. In our experiments, transcellular diffusion appeared extremely unlikely as sulforhodamine 101 (and the other polar tracers in Fig. 2) never appeared/accumulated in the cytoplasm of endothelial cells, i.e. it can hardly have reached the brain via transcellular diffusion. We believe that sulforhodamine 101 cannot cross the membrane of endothelial cells and therefore does not accumulate in their cytoplasm. However, whether it does not accumulate in endothelial cells because it does not enter the cells or whether it enters and then is completely extruded due to MRP1 activity, as pointed out by the reviewer, would in our eyes not make a difference for our conclusion.

How does the lack of perfusion in the brain slice preparation affect barrier properties? Is this possible to test? If not, this should be discussed as one of the disadvantages/unknowns in this system vs. other approaches.

In the ISMICAP model there are two types of perfusions:

a) the typical perfusion of the bath chamber in which the acute brain slices are kept. sACSF is perfused at 2-5 ml/min/chamber volume ~2 ml with a peristaltic pump. This is the standard condition for electrophysiology in slices and slices are known to remain viable for experiments for many hours.

b) we perfuse the vasculature with sACSF. As the volume of the vessels is “microscopic”, the amount of sACSF contained in a glass capillary/pipette (5 µl) is sufficient to keep a constant flow through the impalement pipette and the vasculature for the duration of the experiment (up to 2 h). We did not attempt to modify the perfusion solution in this study.

We see the possibility to vary the capillary perfusion as a great advantage of ISMICAP

and this is discussed in lines 496-512.

Statistics/group numbers:

- If N is 10 or less per group, it is strongly recommended that individual data points be represented on charts (this is done on some, but not all). If N is less than 5 on a given test, the authors should explicitly note this in the text and be cautious in interpretation of significant results.

We had stated the 'n' values for corresponding graphs in the detailed figure legends (for example in Fig. 1d, Fig. 4d&f, Fig. 6b&e). Following the reviewer's suggestion, we now indicate the individual data points on the charts according to the requirements of Nature Communications.

- Statistical terminology is odd: a "dataset" refers to all the data collected from an experiment. The authors should use the term "treatment", not "dataset".

Based on the reviewer's comment, we changed the word 'dataset' to 'groups' in two occurrences (line 949).

- Did the authors use unprotected multiple t tests when performing pairwise comparisons after ANOVA when there were more than two treatments? If so, it is recommended they repeat their pairwise comparisons with adjustment for multiple comparisons.

For experiments where we had more than two treatments, we used Tukey's post *hoc* test, in which we opted for correcting for multiple comparisons using statistical hypothesis testing. We now clearly state this way of statistical testing in the M&M section (lines 950-951).

Reviewer #3 (Remarks to the Author):

One of the limitations that have not been discussed is the intrinsic problem of media perfusion, which removes blood-borne differences from the intraluminal perfusate. This is in particular a problem with pilocarpine: when administered peripherally, pilocarpine exerts its pro-seizure effects by a peripheral mechanism (1-4). This needs to be discussed.

1. Marchi N, Johnson AJ, Puvenna V, Johnson HL, Tierney W, Ghosh C, Cucullo L, Fabene PF, Janigro D.

Modulation of peripheral cytotoxic cells and ictogenesis in a model of seizures. Epilepsia. 2011;52(9):1627-34. Epub 2011/06/02. doi: 10.1111/j.1528-1167.2011.03080.x. PubMed PMID: 21627645; PMCID: PMC3728674.

2. Uva L, Librizzi L, Marchi N, Noe F, Bongiovanni R, Vezzani A, Janigro D, de Curtis M. Acute induction of epileptiform discharges by pilocarpine in the *in vitro* isolated guinea-pig brain requires enhancement of blood-brain barrier permeability. *Neuroscience. 2008;151(1):303-12. Epub 2007/12/18. doi: 10.1016/j.neuroscience.2007.10.037. PubMed PMID: 18082973; PMCID: PMC2774816.*

3. Marchi N, Oby E, Fernandez N, Uva L, de Curtis M, Batra A, Santaguida S, Barnes V, van Boxel A, Najm I, Janigro D. *In vivo* and *in vitro* effects of pilocarpine: relevance to epileptogenesis. *Epilepsia. 2007;48(10):1934-46.*

4. Uva L, Janigro D, de Curtis M. Submillimolar and supramillimolar concentrations of pilocarpine induce, respectively, gamma oscillations and epileptiform activity in the *in vitro* isolated guinea pig brain. *Soc for Neuroscience Annual Meeting. 2006;Atlanta, GA.*

In this study, we combined the *in vivo* pilocarpine animal model of epilepsy with our slice-based ISMICAP approach. Pilocarpine was injected *in vivo* so that it could exert a broad range of changes associated with epileptogenesis, including *status epilepticus*. This may or may not require central and /or peripheral effects but the mice survived as long as pilocarpine was present⁽²⁾. Following 2/4 to 24 h after *status epilepticus* mice were sacrificed and slices prepared for the ISMICAP experiment. To combine *in vivo* animal models with a high-resolution slice approach is a key advantage of ISMICAP. However, we fully agree with the reviewer that for certain questions removing the blood from the vasculature in the slice can be very limiting, e.g. oxygenation of brain tissue from *erythrocytes* cannot be studied with ISMICAP. This limitation is and was clearly mentioned in lines 513-523.

This paper is testimony of technological and manual excellency. I find the approach used extremely appealing and perhaps not too difficult to adopt if someone with an electrophysiological background is available. The results are often very solid, and the overall quality of data is impressive. There are still several odd sentences that require professional editing, but this is easily done.

We would like to thank the reviewer for his appreciation of our study.

The revised manuscript was carefully edited by a native English neuroscientist. If nevertheless odd sentences remain, we would be grateful if they could be pointed out to us.

There are methodological and conceptual issues however, which I describe below. In general, they are related to the translational relevance of the paper. I suggest that the Authors consult with those who developed mannitol or BBB disruption (or others who still use it), and to read with more attention the literature on 3D models and pilocarpine SE. In other words, and exceptional paper with a few non-fatal deficiencies.

Line 111: It is not true that 3D models of the BBB lack perfusion! The first model (3D and perfusion) is from the 1990s (Stanness et al., see also 5, 6). Thus, this sentence in the introduction needs to be modified to acknowledge the state of art in the field. Same for statement line 113: The models cited above have been extensively used to study drug delivery issues e.g., 7. A distinct advantage of this approach is exactly what the Authors try to achieve with pilocarpine: study drug passage in epilepsies. We the model cite above, this is done

directly on human epileptic brain endothelial cells (8,9). Please discuss.

We thank the reviewer for bringing this omission to our attention. We now have deleted the incorrect statements regarding a lack of perfusion in those models (former lines 110 to 114).

Did you record any electrical activity in sclerotic hippocampi from brain resection?

We did not record activity from human hippocampi in this study. We know from multiple previous studies that there is no spontaneous activity in human slices, as it has also been observed in mouse and rat hippocampus. Electrical stimulation of axons or individual cells readily evokes activity of neurons and synapses⁽³⁻⁷⁾.

What part of the hippocampus did you study (DG, CA1, etc.).

As indicated on page 36, we investigated the hippocampal CA1 region.

Shouldn't the sclerotic hippocampus be permeant to gadolinium?

We cannot say whether the human or mouse BBB is permeable to gadolinium as we have not tested it. The MRI appearance of gadolinium may be transient and occur only in the hours following a seizure⁽⁸⁾. In mice, the delay has been suggested to be 48 h, such that this permeability phase would not be visible in our 4 and 24 h time frame⁽⁹⁾. We also would like to point out that while gadolinium has been proven useful within clinical diagnostics and is assumed to indicate BBB damage, it has not yet been shown that it really crosses the BBB. As another possibility, it has been suggested that gadolinium may enter the brain via the cerebrospinal fluid (blood-CSF-barrier) in mice and man (i.e. Plexus choroideus)^(10, 11).

Please show the contrast MRI from the same hippocampus to convince the reader that here was no leakage in vivo, or remove this section from the paper: The "sturdiness" of the vessels may be due to the sclerotic nature of the tissue.

Our slice experiments with the ISMICAP approach tested properties of the human BBB. Therefore, they stand for themselves and do not need "approval" by a contrast MRI from the patient. We do not understand why we should remove this data if we cannot add MRIs. Both approaches have their own value and may also measure slightly different processes (see paragraph above), such that neither of them validates or invalidates the other. We have carefully checked our text and feel that we do not draw unjustified or too strong conclusions from our human slice experiment nor do we claim that any previous studies were incorrect. We conclude that we could not detect extravasation of albumin under our conditions, which is exactly what we observed.

A reviewer noted that surface not volume was reported for the ROI. This is not at all trivial as the rebuttal suggests in particular if drug passage is being measured. In the equation $Permeability = P_{app} \times S$, the term P_{app} contains the volumes of the luminal and abluminal tracer (presented as concentrations). Please address

As we indicated in our previous rebuttal letter, we quantified the surface area of ECs instead of the volume (Fig. 3) as an accurate estimate of the volume of an EC is difficult to obtain. Furthermore, the surface area, and not the volume, also is the main determinant of the apparent permeability coefficient (P_{app}) mentioned by the reviewer⁽¹²⁾, which is expressed as: $P_{app} = dQ/(dt \times A \times C_0)$, where:

dQ/dt is the flux per time

and A is the surface area,

C_0 is the initial solute concentration on the luminal side.

There is a great deal of confusion as to the experiments with mannitol. The concentration of use for BBB disruption is not 15%, which is used to reduce brain swelling. The molarity of the solution used by Neuwelt and others is three times the osmolarity of blood, injected intracarotid. The Authors need to explain their findings with a concentration that in fact is used clinically to improve BBB function. What were the effects of mannitol on vessel diameter?

Previous work in rodents and humans has shown that injections of 25% w/v mannitol for typically up to 30 sec (equivalent to 1.4 M) *reversibly open* the BBB (it does not “damage” the BBB) by increasing the volume in the vasculature due to osmotic flux of water from the brain⁽¹³⁾.

In our study, we used a mannitol concentration of 15% (0.82 M) as a higher concentration precipitated at 18°C laboratory temperature. With 15%, we observed a similar opening of the BBB as detected with 25% and the reviewer seems to wonder how this is possible.

To compare the concentrations, it is necessary to understand that with the ISMICAP approach we directly define the concentration of mannitol in the blood vessel **by perfusion**, we achieve a 15% mannitol concentration intravascularly. In contrast, in previous *in vivo* work a 25% mannitol solution was **injected**, meaning that this solution was added to the blood stream via a catheter. This approach results in a dilution of the injected substance by the circulating blood volume. In fact, previous publications cannot accurately determine the concentration of mannitol reached in the vasculature. Considering that an adult rat has a blood volume of ~20 ml and that previous studies injected ~3.5 ml of a 25% mannitol solution, a strong effective dilution factor (of the peak intravascular concentration of mannitol) of at least 2 is very likely⁽¹⁴⁾. Therefore, it is to be expected that we achieve a strong BBB opening with an effective 15% mannitol

concentration. This example demonstrates one of the key features of ISMICAP for quantitative studies.

We are grateful for the suggestion to analyze vessel diameter which can be done precisely with ISMICAP. For each vessel, we determined its diameter after 5 min and 30 min of perfusion. The capillaries' diameter was increased by mannitol by ~8.4% translating into an increase in cross sectional area of ~17.5%. In contrast, the diameter of venules remained unaffected by mannitol, suggesting that mannitol primarily acts on the tightness of capillaries. We now added this analysis to the manuscript (lines 345-351).

The concentration of probenecid is very high and there is concern on the vehicle effects without the drug. Any issue here worth discussing? What were the effects of vehicle alone? Also, you report 1 mM and 0.6 mM. Which one was used? And if both were used, why the different concentrations?

Probenecid was prepared in DMSO as 1 M stock solution. Therefore, the final DMSO concentrations in the 0.6 mM and 1 mM probenecid solution in sACSF were 0.06% and 0.1%, respectively. Based on our experiments, DMSO concentrations up to 1% in ACSF did not significantly alter BBB permeability. In the calcein assay in Fig. 4, a concentration of 1 mM probenecid was used, while we applied a concentration of 0.6 mM in the pilocarpine SE experiments (Fig. 7). Both concentrations are in the upper range of those used previously to establish the complete inhibition of ABCC1 transporters ⁽¹⁵⁻¹⁹⁾.

We conducted a concentration-response experiment to determine the saturation and sub-saturation concentrations of probenecid (see graph). Both concentrations of probenecid (0.6 and 1 mM) resulted in a comparable calcein accumulation. However, preparing 1 mM probenecid in sACSF was challenging due to solubility issues. Therefore, we opted for using 0.6 mM in later experiments.

Question 1: Is there a readout of the pipette perfusion pressure, beginning and end?

Initially, we tightly controlled the pressure applied to the back of the perfusion pipette and kept it at 100 mmHg. However, we later figured out that applying pressure with an insulin syringe (which we use for patch clamp, too) such that blood cells are flushed

out and the collapse of vessel is reversed is as effective and more convenient.

Question 2: Would it be possible to add leukocytes to the perfusate? See comment above.

In principle, the perfused solution can be customized to suit the purpose of the study. However, the perfused components should be able to pass freely through the pipette tip. For our experiments, we chose a small diameter of the pipette opening of about 3 μm which is optimized for successful impalement. To inject cells, the diameter needs to be beveled back to approximately 20 μm . The bevel process will also sharpen those pipettes and impalement should be possible, probably best targeting larger venules. We have not yet tried this approach but can imagine that optical monitoring of BBB traverse by leukocytes may be exciting .

References:

1. McCaffrey, G., Staats, W. D., et al. P-glycoprotein trafficking at the blood–brain barrier altered by peripheral inflammatory hyperalgesia. **Journal of neurochemistry**. 2012;122(5):962-75.
2. Mazzuferi, M., Kumar, G., et al. Rapid epileptogenesis in the mouse pilocarpine model: video-EEG, pharmacokinetic and histopathological characterization. **Experimental neurology**. 2012;238(2):156-67.
3. Dietrich, D., Clusmann, H., et al. Two electrophysiologically distinct types of granule cells in epileptic human hippocampus. **Neuroscience**. 1999;90(4):1197-206.
4. Dietrich, D., Kral, T., et al. Reduced function of L-AP4-sensitive metabotropic glutamate receptors in human epileptic sclerotic hippocampus. **Eur J Neurosci**. 1999;11(3):1109-13.
5. Dietrich, D., Kral, T., et al. Presynaptic group II metabotropic glutamate receptors reduce stimulated and spontaneous transmitter release in human dentate gyrus. **Neuropharmacology**. 2002;42(3):297-305.
6. Selke, K., Müller, A., et al. Firing pattern and calbindin-D28k content of human epileptic granule cells. **Brain Res**. 2006;1120(1):191-201.
7. Surges, R., Kukley, M., et al. Hyperpolarization-activated cation current Ih of dentate gyrus granule cells is upregulated in human and rat temporal lobe epilepsy. **Biochem Biophys Res Commun**. 2012;420(1):156-60.
8. Rüber, T., David, B., et al. Evidence for peri-ictal blood–brain barrier dysfunction in patients with epilepsy. **Brain**. 2018;141(10):2952-65.
9. Boux, F., Forbes, F., et al. Neurovascular multiparametric MRI defines epileptogenic and seizure propagation regions in experimental mesiotemporal lobe epilepsy. **Epilepsia**. 2021;62(5):1244-55.
10. Jost, G., Lenhard, D. C., et al. Signal Increase on Unenhanced T1-Weighted Images in the Rat Brain After Repeated, Extended Doses of Gadolinium-Based Contrast Agents: Comparison of Linear and Macrocyclic Agents. **Invest Radiol**. 2016;51(2):83-9.
11. Mamourian, A. C., Hoopes, P. J., et al. Visualization of intravenously administered contrast material in the CSF on fluid-attenuated inversion-recovery MR images: an in vitro and animal-model investigation. **AJNR Am J Neuroradiol**. 2000;21(1):105-11.
12. Schoenwald, R. D., Huang, H. S. Corneal penetration behavior of β -blocking agents I: physicochemical factors. **Journal of pharmaceutical sciences**. 1983;72(11):1266-72.
13. Rapoport, S. I. Osmotic opening of the blood-brain barrier: principles, mechanism, and

- therapeutic applications. **Cellular and molecular neurobiology**. 2000;20(2):217-30.
14. Rapoport, S. I., Fredericks, W., et al. Quantitative aspects of reversible osmotic opening of the blood-brain barrier. **American Journal of Physiology-Regulatory, Integrative and Comparative Physiology**. 1980;238(5):R421-R31.
 15. Flores, K., Manautou, J. E., et al. Gender-specific expression of ATP-binding cassette (Abc) transporters and cytoprotective genes in mouse choroid plexus. **Toxicology**. 2017;386:84-92.
 16. Gollapudi, S., Kim, C. H., et al. Probenecid reverses multidrug resistance in multidrug resistance-associated protein-overexpressing HL60/AR and H69/AR cells but not in P-glycoprotein-overexpressing HL60/Tax and P388/ADR cells. **Cancer chemotherapy and pharmacology**. 1997;40(2):150-8.
 17. Cisternino, S., Rousselle, C., et al. Apparent lack of Mrp1-mediated efflux at the luminal side of mouse blood-brain barrier endothelial cells. **Pharmaceutical research**. 2003;20(6):904-9.
 18. Neuhaus, W., Stessl, M., et al. Blood–brain barrier cell line PBMEC/C1-2 possesses functionally active P-glycoprotein. **Neuroscience letters**. 2010;469(2):224-8.
 19. Gedeon, C., Behravan, J., et al. Transport of glyburide by placental ABC transporters: implications in fetal drug exposure. **Placenta**. 2006;27(11-12):1096-102.

REVIEWERS' COMMENTS

Reviewer #2 (Remarks to the Author):

The authors adequately addressed my comments.

Reviewer #3 (Remarks to the Author):

All comments were addressed adequately.